# A volcanic hazard demonstration exercise to assess and mitigate the impacts of volcanic ash clouds on civil and military aviation

Marcus Hirtl[1,16], Delia Arnold[1,2], Rocio Baro[1], Hugues Brenot[3], Mauro Coltelli[4], Kurt Eschbacher[5], Helmut Hard-Stremayer[6], Florian Lipok[7], Christian Maurer[1], Dieter Meinhard[7], Lucia Mona[8], Marie D. Mulder[1], Nikolaos Papagiannopoulos[8], Michael Pernsteiner[9], Matthieu Plu[10], Lennart Robertson[11], Carl-Herbert Rokitansky[5], Barbara Scherllin-Pirscher[1], Klaus Sievers[12], Mikhail Sofiev[13], Wim Som de Cerff[14], Martin Steinheimer[15], Martin Stuefer[16], Nicolas Theys[3], Andreas Uppstu[13], Saskia Wagenaar[14], Roland Winkler[15], Gerhard Wotawa[1], Fritz Zobl[5], Raimund Zopp[17]

[1]Zentralanstalt für Meteorologie und Geodynamik, Vienna, A-1190, Austria
[2]Arnold Scientific Consulting, Manresa, 08242, Spain
[3]Support to Aviation Control Service, BIRA-IASB, Brussels, B-1180, Belgium
[4]Osservatorio Etneo, Istituto Nazionale di Geofisica e Vulcanologia, Catania, 95125, Italy
[5]Department of Computer Sciences, University of Salzburg, Salzburg, 5020, Austria
[6]Kommando Streitkräfte/FachstabLu/J3(Lu), RefLtr Luftraumüberwachung, St. Johann im Pongau/Betriebsstelle Plankenau, 5600, Austria
[7]Brimatech Services GmbH, Vienna, A-1030, Austria
[8]Consiglio Nazionale delle Ricerche, Istituto di Metodologie per l'Analisi Ambientale (CNR-IMAA), Tito Scalo (PZ), 85050, Italy
[9]Joint Forces Command / Airstaff, Schwarzenbergkaserne, Wals, 5071, Austria
[10]CNRM, Université de Toulouse, Météo-France, CNRS, Toulouse, 31057, France
[11]Swedish Meteorological and Hydrological Institute, Norrkoping, SE-601 76, Sweden
[12]Klaus Sievers Aviation Weather, Lenggries, 83661, Germany
[13]Atmospheric Composition Research, FMI, Helsinki, FI-00101, Finland
[14]R&D Satellite Observations, KNMI, De Bilt, 3731 GK, Netherlands
[15]Austro Control GmbH, Vienna Int. Airport, Schwechat, 1300, Austria
[16]Geophysical Institute, University of Alaska Fairbanks, Fairbanks, AK 99775, USA
[17]Flightkeys, Vienna, A-1070, Austria

*Correspondence to*: Marcus Hirtl (marcus.hirtl@zamg.ac.at)

**Abstract.** Volcanic eruptions comprise an important airborne hazard for aviation. Although significant events are rare, e.g. compared to the threat of thunderstorms, they have a very high impact. The current state of tools and abilities to mitigate aviation hazards associated with an assumed volcanic cloud was tested within an international demonstration exercise. Experts in the field assembled at the Schwarzenberg barracks in Salzburg, Austria, in order to simulate the sequence of procedures for the volcanic case scenario of an artificial eruption of the Etna volcano in Italy. The scope of the exercise ranged from the detection (based on artificial observations) of the assumed event to the issuance of early warnings. Volcanic emission concentration charts were generated applying modern ensemble techniques. The exercise products provided an important basis for decision making for aviation traffic management during a volcanic eruption crisis. By integrating the available wealth of data, observations and modelling results, directly into a widely used flight planning software, it was demonstrated that route optimization measures could be implemented effectively. With timely and rather precise warnings available, the new tools and

processes tested during the exercise demonstrated vividly that a vast majority of flights could be conducted despite a volcanic plume widely dispersed within a high-traffic airspace over Europe. The resulting number of flight cancellations was minimal.

## 1 Introduction

Aviation is nowadays one of the most critical ways of transport and even short interruptions in flight schedules can result in major economic loses. Volcanic eruptions can have a significant impact on aviation (e.g. Bolic and Sivcev, 2011; Guffanti et al., 2010a; Albersheim and Guffanti, 2009; Brechan, 2010) not only in the vicinity of an erupting volcano, but also far from the volcano. Especially fine ash typically defined as ash particles with diameters below 63 µm, can be transported over very long distances (Durant et al., 2012; Prata et al., 2007).

Volcanic ash, particularly fine ash, may represent a significant danger to aviation, as volcanic ash particles can deteriorate forward facing surfaces and metal components. Although ash would not destroy the engine, high concentrations of volcanic ash particles are a problem as they melt due to lower melting temperature than the temperature within engine and stick to turbine surfaces that can lead to a blocking of the aerodynamic throat area (Clarkson et al., 2016). Aircraft engines may sustain sudden power loss (Bolic and Sivcev, 2011) or even complete engine failure (ICAO, 2007). Volcanic ash generally may cause a variety of additional damages inside and outside an aircraft. The silicate particles of an ash cloud scratch cockpit screens or grind down various parts of the aircraft, such as the leading edge of the wings, or tail parts instantly impacting aerodynamics in flight. In addition, the pitot tubes, which are essential for speed and altitude measurement, can be clogged. Volcanic ash may cause blockages of the air filters and contamination of the inside cabin air.

The first incident concerning engine flameout due to high-level (i.e. at cruise altitudes) volcanic ash clouds happened on June 24, 1982. A Boeing 747-200 of British Airways lost power on all four engines when the aircraft entered at 11000 m in an ash-contaminated airspace due to the eruption of volcano Mount Galunggung, Indonesia (Johnson and Casdevall, 1994). During the ensuing sixteen minutes, the aircraft descended without power from 11000 m to 3600 m; in last minute, the pilots were able to restart three of the four engines (ICAO, 2007) and therefore regained control without costs to human lives.

Other incidents include a KLM Boeing 747, on December 15, 1989, which was for more than four minutes powerless as the plane flew into a dense ash contaminated airspace 170 km north of Anchorage, Alaska (Przedpelski and Casadevall, 1994). The ash-cloud originated from Redoubt, at a distance of 240 km from the aircraft. Notable, both incidents occurred during night, the volcanic plumes were invisible and no warnings were available to the pilots during these days. A more recent incident happened during July and August of 2008, the Kasatochi (Guffanti et al., 2010b) volcano in Alaska's Aleutian Islands chain, erupted, causing widespread impacts on aviation operations. The volcanic ash and gas were ejected up to the stratosphere and dispersed into major North American jet airways.

Especially the first events, with an impact on aviation, were the starting point of the development of guidelines, legislations and specific products for aviation. In 1991, the International Civil Aviation Organisation (ICAO) and the World Meteorological Organization (WMO) decided to setup Volcanic Ash Advisory Centres (VAACs) to monitor and forecast the dispersion of

fine ash in the atmosphere. In Europe, the London VAAC (UK Met Office) and the Toulouse VAAC (Météo-France) ensure this responsibility to operate a 24/7 service. There are many contributors to the overall volcanic risk mitigation system such as, Air Navigation Service Providers (ANSP) including Aeronautical Information Services and Air Traffic Flow Management (AFTM) Units, Meteorological Service providers including Meteorological Watch Offices (MWOs), Volcano Observatories.

Original Equipment Manufacturers (OEM) are required to establish acceptable susceptibility means of engine features to the effects of volcanic clouds. These should include a combination of experience, studies, analysis, and/or testing of parts, sub-assemblies or engines. Specific aircraft and engine type certificates, supplemental type certificates (STC) and parts manufacturer approvals (PMA) have been developed to consider the possibility of volcanic particle exposure. Their cooperation in supplying governments, operators and Civil Aviation Authorities (CAAs) with the information necessary to

support the pre-flight, in-flight and post-flight decision-making process is essential for continuous safe operations.

The eruption of the Eyjafjallajökull volcano in Iceland in April and May 2010 caused a major crisis situation especially for aviation. The crisis happened despite some decades of experience and new technologies available to detect volcanic ash clouds in real time. Satellite remote sensing networks were available (although e.g. with less quality and resolution than nowadays) and advanced communication channels existed in 2010. Nevertheless, major interruptions and a significant collapse of the air

traffic system over Europe could not be prevented for days (Bolic and Sivcev, 2011). The Eyjafjallajökull 2010 crisis (e.g. Stohl et al., 2011, Bolic and Sivcev, 2012, Gudmundsson et al., 2012) highlighted the societal demand for unaffected mobility, as well as aviation vulnerability to natural hazards. Estimated seven million passengers were left stranded, with the effects of the closing of airports extending to trade, business, and general production. Although, there were no casualties, the economic impact was enormous (IATA, 2010). Especially during the first days after the eruption, from April 15 until April 22, 2010,

104000 flights were cancelled (Alexander, 2013), which comprised 48 % of the expected traffic during those days.

Since the Eyjafjallajökull outbreak in 2010 in Iceland and the resulting closing of wide areas of the European airspace for days, changes were made to the standards and recommended practices of aviation in the case of volcanic eruptions. Until this event, the maxim was "Avoid Visible Ash" as answer to the flights of two Boeings 747 into ash clouds in the 1980s (Johnson and Casadevall, 1994), which triggered the establishment of the International Airways Volcano Watch (IAVW) and respective

ICAO procedures and guidelines. This rule is still valid for the conduct of flights today. Within this context, the Volcanic Ash Contingency Plan (VACP, Edition 2.0.0 – July, 2016) grants airspace users the decision whether to fly or not to fly based on their Safety Risk Assessment accepted by the Civil Aviation Authority of the State of registration. This includes the decision about operation in airspace where volcanic ash is present or forecast. Consequently, most countries in Europe do not close their airspace as a default procedure in the event of a volcanic eruption.

As a consequence of the Eyjafjallajökull eruption, VAACs London and Toulouse are providing volcanic ash concentration charts to support the VACP. These charts predict the location of a quantitative mass of ash per unit volume and are provided for three contamination levels:

- "Low contamination" volcanic ash mass concentration less than or equal to 2 mg/m³
- "Medium contamination" volcanic ash mass concentration greater than 2 mg/m³ and less than 4 mg/m³

- "High contamination" volcanic ash mass concentration greater than or equal to 4 mg/m³

The Eyjafjallajökull eruption showed that there is still a significant gap in the Europe-wide availability of real-time hazard mitigation data and tools. The 2010 event revealed a significant lack of volcanic monitoring information for airborne hazards, and informative and crucial data describing "what, where, and how much" were missing. Within the framework of the H-2020 research project EUNADICS-AV ("European Natural Disaster Coordination and Information System for Aviation", 2016-2019) funded by the European Commission, a network of experts was established, with the aim to provide the relevant data (observations and models) during situations when aviation is affected by airborne hazards (e.g. volcanic ash and $SO_2$, desert dust, wildfires and nuclear accidents). To enable all stakeholders in aviation to obtain fast, coherent, and consistent information, all data is collected and visualized on a dedicated platform (see section 3.1.2). In order to implement efficient data and information exchange, interfaces between various data sources from observational networks, dispersion modelling applications and flight planning software were developed. This included the linking of dispersion models with flight trajectory models that consider cost-based re-routing of flights (with respect to fuel and maintenance costs, see section 5.3).

All developments were tested and evaluated at an international demonstration exercise conducted at the barracks of the Austrian Armed Forces (AAF) in Salzburg in March 2019. The goal of the demonstration exercise was to simulate the different phases of the event, expert contributions and decision procedures during a volcanic eruption and a nuclear crisis. This paper focuses on the volcanic scenario.

The following sections describe the motivation and organization of the exercise, the assumed scenario of the eruptive event, available data and tools and results and conclusions derived by the participating collaborators. The case scenario starts with the early detection of the fictitious eruption of the Etna volcano, Italy, and subsequent early warnings were developed. There were observations from different sources available (e.g. satellite, LIDAR, in situ), which were used to analyse the hazard situation. The observational data were combined with state-of-the art simulation models to determine the source terms of the harmful substances and to further refine the analysis. These results were then fed into flight trajectory and air traffic simulation models.

## 2 International exercises

In the field of dispersion modelling and the related impacts (e.g. on aviation), different exercises have been conducted in the past and some of them, e.g. the VOLCEX (VOLcanic ash Contingency EXercise; Sivcev, 2011; Dopagne, 2011) are repeated on a regular basis to test how procedures perform in real time situations. The exercises are means to test communication networks and data exchange capabilities between the involved centres and groups. Important goals of such exercises are also checking the distribution of responsibilities and how seamless and coordinated necessary tasks are completed.

For volcanic ash, the annual European VOLCEX is the most important exercise with respect to volcanic ash and the impacts on aviation. The VOLCEX exercise involves the participation of VAACs, different airlines, ANSPs, as well as different

regulators and other crisis coordination cells. The objective of the exercise is to provide an opportunity for each individual state to test the effectiveness of their national crisis procedures, and for all the participants, their local volcanic ash contingency plans and procedures. It is designed to test the operational capabilities and speed of all players involved in the industry (e.g. airlines) that could be affected by a volcanic eruption in EU airspace. During the exercise, the crisis coordination between the various stakeholders via the European Aviation Crisis Coordination Cell (EACCC) and the Aircraft Operator Crisis Coordination Cell (AOCCC) is evaluated.

The EUNADICS-AV demonstration exercise was unique with respect to the abovementioned activities. The exercise did not only comprise all the items and timescales for a potential event relevant for aviation, but also looked at it from a research-oriented perspective. Innovative procedures, data and products were tested in a simulated environment. Such a complete and comprehensive exercise demonstrated the applicability and feasibility of these innovative solutions into the aviation sector, produced by the research and operational capabilities of the EUNADICS-AV partnership. The exercise also demonstrated vast opportunities to support and complement the VAAC activities in the future e.g. by providing relevant observations, early warnings, source terms and analysis fields via a dedicated platform. This paper continues with describing the several activities during the exercise with focus on the volcanic test case and the lessons learnt from this multidisciplinary exercise.

## 3 Overview of the EUNADICS-AV demonstration exercise set-up

### 3.1 General approach

The exercise took place at the military facilities at the Schwarzenberg barracks from March 3 to March 8 of 2019. The floor plan and location of the different exercise cells (Fig. 1) show the organisational aspects of the exercise and how the interaction among cells was facilitated. Each cell was in charge of one of the pre-defined actions within a crisis situation (see Table 1 for details).

Starting from (1) the detection (based on artificial observations) of the hazardous event and (2) the declaration of early warnings, (3) observations from different sources (e.g. satellite, LIDAR, in situ) were used to analyse the situation and were furthermore combined with model to determine the source terms and to refine the analysis (4). These results were then used to (5) cost-efficiently re-route airplanes. Every step of the whole procedure was executed with a demonstration of which data would be used in a real event and how the procedures and dependencies would take place (Fig. 3).

During the demonstration exercise, the cells presented the relevant data and impact on aviation during four phases of the event defined as:

- **Pre-alert**: A notification of an event is received, which may lead to a possible major disruption of ATM (Air Traffic Management), requiring activation of an operational reaction chain.
- **Disruption**: Major disruption that impacts ATM operations and which may escalate to a crisis.

- **Crisis**: State of inability to provide air navigation service at required level resulting in a major loss of capacity, or a major imbalance between capacity and demand, or a major failure in the information flow following an unusual and unforeseen situation.

- **Recovery**: The operation will go back to normal and an evaluation of the impact will be finalized.

Each cell was equipped with technical support, e.g., computers and monitors to demonstrate its role in the operational sequence of procedures and actions. Main results could be projected from each cell to various big screens in the hall (see Fig. 2).

Based on the observations and modelling data provided (see section 4) by the "scientific cells" (cell 2 to 6), the impact on aviation was simulated and depicted by the "aviation and stakeholder cells" (cell 7 to 15). These data allowed the ANSPs to

10 release special aviation advisories (e.g. Significant Meteorological Information - SIGMET). Further impacts were due to re-routing and cancellation of flights by flight trajectory modellers (see section 5.3) as well as respective procedures invoked on the military side and participating airports and ministries. For the evaluation of the impacts on the air traffic, the air traffic was simulated by using the NAVSIM/USBGSim (Rokitansky, 2009) simulator ~~and was analysed by using Key Performance Indicators (KPIs)~~. A newly developed cost model was used in the framework of an airline network balance tool (Flightkeys

5D) to cost-efficiently re-route flights affected by the disaster event (see section 5.3). Cell-specific EUNADICS-AV developments and impacts on ATM were shown for both, the civil and the military roles within each ATM phase.

The most important tasks in the preparation phase of the demonstration exercise, were to establish working practices and interfaces between the broader natural hazard science communities on one side and the more application-oriented aviation community on the other side. The latter was mostly represented by flight trajectory modellers and military aviation experts.

The emphasize at the EUNADICS-AV exercise was more on the scientific part and not on regular procedures, which are e.g. tested during the VOLCEX exercise where the crisis coordination between the various stakeholders ~~through EACCC (the European Aviation Crisis Coordination Cell) and the Aircraft Operator Crisis Coordination Cell (AOCCC)~~ is tested and evaluated. The intense preparatory work ahead of the demonstration exercise managed to bridge existing gaps. Experts such as natural hazard scientists to flight managers and pilots collaborated in an unprecedented fashion. This effectively was the

spirit of the EUNADICS-AV project put into practice.

### 3.1.1 The volcanic eruption scenario

The exercise scenario was designed assuming a fictitious eruption of the Etna volcano in Sicily, Italy. The aim was to simulate the onset of the eruption during an episode with large scale weather patterns that led to a transport of ash from Sicily towards central Europe over a couple of days, with a wide spread ash cloud with concentration levels above the relevant thresholds

(e.g. above 2 mg m$^{-3}$ at any vertical sub-column, see also section 1). A period lasting one week during April 2018 was chosen. The fictitious Etna eruption started on April 18, 2018, at 12:00 UTC with a plume height of 12 km above the vent and a constant ash emission of 198 t/s. The eruption should last until April 20, 00:00 UTC. The plume height and emission rate were

further assumed to be 10 km and 116 t/s until April 22, 00:00 UTC. The following days until April 25 were considered as the "recovery phase", when no relevant ash source was present over Europe anymore (see also section 4.3).

A fictitious event was chosen because there was no real eruption in the past that fulfilled the requirements which were defined for the exercise, e.g. that the volcanic ash cloud spread over Europe with certain threshold exceedances of volcanic ash concentrations over selected regions (e.g. over Austria where ANSP have to deliver specific products).

### 3.1.2 Data sharing and visualization

All data sets (artificial observations and model data, see section 4) used for the demonstration exercise were accessible and visualized. The information flow is depicted in Fig. 4. The various data products (satellite and ground-based observations, modelling data) that were requested by the wide range of users, had many different data sources, data types and formats, projections, sampling time intervals, and coverage.

The EUNADICS-AV project made use of existing data channels and protocols to provide a harmonised and easily accessible portal (see Fig. 5 for example) for all the different types of information, including observations or modelling results. The portal allowed the participants of the exercise to explore the event and the products available anytime during each of the four phases with a graphical user interface.

## 4 Datasets used for the demonstration exercise

### 4.1 Artificial observations

As the considered Etna scenario was artificial, the observational data sets were generated based on model simulations. The task of creating the artificial observations splits to two steps: (i) simulate the evolution of the artificial volcanic plume, (ii) simulate the fingerprint of this plume for different types of observation devices. An artificial eruption scenario was chosen, because in the recent past there was no real eruption of the Etna that would have resulted in long range transports of ash over central Europe with concentration values over certain thresholds.

### 4.1.1 Simulations of the artificial plume evolution

Having the parameters of the artificial eruption decided, the SILAM (System for Integrated modeLling of Atmospheric composition) modelling system of FMI (http://silam.fmi.fi, Sofiev et al, 2015) was run over the European domain simulating the dispersion of the emitted masses in the atmosphere (see example in Fig. 6, left).

The main challenge at this stage was to simulate not only the main dispersion but also account for incompleteness of our knowledge of the atmosphere, e.g. its dynamics during these days, actual wind direction and speed, etc.. To account for this uncertainty, the SILAM model was not run with meteorological data from the operational weather models but rather from the reanalysis dataset ERA5 (Hersbach and Dee, 2016) of the European Centre of Medium-Range Weather Forecast (ECMWF). ERA5 was produced by assimilating vast amounts of historical observations into a numerical model and provides therefore

high-quality meteorological data on a global scale, the final data set will extend back to 1950. Compared to the operational weather predictions, the re-analysis has noticeably more extensive assimilation capabilities (see summary for the previous version of the ECMWF reanalysis ERA-Interim in (Dee et al., 2011)) and thus can be considered as a more accurate representation of the actual meteorological conditions than the operational forecasts. All other simulations with SILAM and
other models used only operational forecasts for the corresponding period.

### 4.1.2 Generation of artificial observations from SILAM simulations

The output of the plume dispersion simulations consisted of the 4-D distribution of the ash concentrations. To obtain the artificial observations, we applied the corresponding observation operators, which "observed" the concentration distribution as if the corresponding device would have done it. For instance, for the in-situ sampling observations, the concentrations near-
surface were extracted at the locations of the stations. For EARLINET (European Aerosol Research Lidar Network; Pappalardo et al., 2014; www.earlinet.org) LIDAR, the vertical profiles of concentrations over the LIDAR locations were convoluted with their sensitivity. For satellites, the averaging kernels of the instruments were convoluted with the vertical profiles of concentrations along the satellite trajectory, etc. We also took into account the actual weather impact and instrument specifics: as shown in the example in Fig. 6 (right), IASI (Infrared Atmospheric Sounding Interferometer, e.g. Karagulian, 2010)
instrument cannot observe in cloudy conditions and cannot retrieve very thick ash layers. At the final stage, the artificial plume retrievals were summed-up with the actual satellite data for the corresponding orbits.

### 4.2 The Early Warning System (EWS)

The purpose of the EWS is to provide near real-time (NRT) observational data in case of the detection of an airborne hazard. Subsequently, it provides centralized information for NRT monitoring, to contribute to the improvement of the analysis and
forecasts of volcanic ash plumes.

Observational data (satellite, ground based and airborne remote sensing and in-situ) play a significant role to determine the 4D distribution of the ash cloud. For the demonstration exercise, several ground- and space-based observations (synthetic) were chosen to facilitate the detection of the event (see in sections 4.2.1 to 4.2.3) and to assess the current extent and location of the dispersed ash cloud.
During the event, alerts created from these synthetic observations were delivered to the different cells to trigger their respective actions. As in a quasi-operative mode, with NRT products derived from the available monitoring networks, appropriate information from notifications of the hazardous event were provided.

### 4.2.1 Volcano observatory Sicily

The Etna Volcano Observatory of the Italian National Institute of Geophysics and Volcanology (INGV) produced five VONA
(Volcanic Observatory Notice for Aviation) messages for the artificial Etna case. The first VONA message was YELLOW, indicating that the volcano showed signs of elevated unrest above its background level (6:00 UTC on April 18, 2018). Then,

10 minutes after the start of Etna eruption (at 12:00 on April 18, 2018), a RED message informed that lava fountaining started from the central crater summit vent, with large ash emission occurring up to 12000 m above the sea level (a.s.l.) and the ash plume moving towards the north. At 00:10 UTC on April 20, 2018, a second RED message was provided. This message confirmed the on-going eruption, with lower plume heights up to 10000 m (instead of 12000 m a.s.l.). Just after 00:00 UTC
on April 23, 2018, an ORANGE message indicated the end of lava fountaining and ash emission. Next, a GREEN message followed, with the announcement that the volcanic activity has ceased (volcano reverts to its normal/non-eruptive state) and that no ash cloud was produced anymore.

### 4.2.2 Synthetic EARLINET/ACTRIS data

The European Aerosol Research Lidar Network (EARLINET; Pappalardo et al., 2014; www.earlinet.org), established in 2000,
provides aerosol profiling data on a continental scale. EARLINET is part of the Aerosols, Clouds, and Trace gases Research InfraStructure (ACTRIS; www.actris.eu).

Within the EUNADICS-AV project, an EWS was designed that relies explicitly on EARLINET aerosol observations. This product is based on the possibilities offered by the EARLINET Single Calculus Chain (SCC; D'Amico et al., 2015) for the NRT data processing and the generation of tailored products network-wide. The calibrated high-resolution LIDAR data serve
as the basis for the alert delivery (Baars et al., 2017). This EWS provides only qualitative and not quantitative information thus the EARLINET EWS represents a warning system rather than a tool for decision makers. The demonstration exercise was the first occasion for which the EARLINET EWS was tested.

The EARLINET products for the demonstration exercise consisted of snap-shots of the LIDAR signals and of the EWS plot. Figure 7 shows an example of the particle backscatter coefficient (with the cloud fraction removed and when not) for the
EARLINET station in Barcelona. Special attention should be given to the particle backscatter coefficient values reported in the figures as the values are unrealistically high and most likely would attenuate the LIDAR laser beam.

The simulated data refers only to volcanic ash and the depolarization information was not incorporated in the alert delivery method. Figure 8 presents an example of the attenuated backscatter coefficient and the corresponding alert for aviation for the EARLINET Leipzig station. The ash layer first appears at around 13 km on April 22, 05:00 UTC, with high values reaching
the ground the next day. The ash cloud is not visible anymore over the Leipzig EARLINET site on April 24.

### 4.2.3 Synthetic satellite data simulated for IASI and MODIS

Satellite observations from 2 types of sensors (IASI and the Moderate-resolution Imaging Spectroradiometer (MODIS)) were considered for generating alert products for the exercise. Clerbaux et al. (2009) and Levy et al. (2015) gave descriptions of the IASI and MODIS instruments respectively. Synthetic ash products from two IASI sensors (onboard MetOp-A and MetOp-B;
Clarisse et al., 2010) and aerosol optical depth (AOD) from two MODIS sensors (onboard Aqua and Terra) were used.

After the detection of the aerosol/ash cloud at the Nicolosi and Catania EARLINET stations, the first selective detection of ash from satellite was created for IASI-A (Fig. 9, left) few ash pixels at 20:10 UTC on April 18, 2018). A first partial detection

was created for 08:28 UTC on April 19, 2018, followed by a global detection of the ash plume by IASI-B (Fig. 9, right) at 09:27 UTC on April 19, 2018. Following each of these synthetic detections a warning was issued with a time delay of about 1.5 hours.

Synthetic observations of the ash plume (AOD anomalies) were created for the MODIS instruments aboard of the Terra and Aqua satellites in similar fashion as for IASI-A and B. The MODIS detections were selected with respective daily timestamps of 10:19 and 11:58 UTC. Warnings were delivered to the aviation and military cells in support of their decision, actions and tasks. The warnings were also used to provide simulation start times for the dispersion models, improving the capability for achieving advanced analysis and forecasts of Etna's ash cloud.

### 4.3 Model ensemble

The following models provided the concentration distribution for the whole period using the pre-defined source term: MATCH (Multi-scale Atmospheric Transport and Chemistry, Robertson et al., 1999), MOCAGE (Modèle de Chimie Atmosphérique de Grande Echelle, Guth et al., 2016), SILAM (Sofiev et al., 2015), FLEXPART (FLEXible PARTicle dispersion model, Stohl et al., 2005) and WRF-Chem (Weather Research and Forecasting (WRF) model coupled with Chemistry, Grell et al., 2005; Stuefer et al., 2013). Although the EUNADICS-AV partnership used a reduced number of models, this approach accounts for an ensemble of multiple models. The 4D-model values were interpolated to a common grid, which allowed to calculate a mini-ensemble and percentiles indicating the model uncertainty for the considered time step and location. For the demonstration exercise, the 75th percentile ash concentration level was used, which corresponds to ash concentrations below 75 % of the modelled outputs. Using this approach was a slightly conservative compromise as the median (50th percentile) is the most probable scenario. Using the 75th percentile means that the regions that lie above a certain threshold are larger than for the median.

Figure 10 shows the dispersion of the volcanic ash cloud over several days at a selected layer (FL275). Note that the model data were produced in advance of the demonstration exercise. The modelled results were used as baseline for the aviation related tasks. The modelled data were interpolated to 13 flight levels, visualized on the EUNADICS-AV portal and also imported into other applications like flight planning software and Visual Weather (https://www.iblsoft.com/products/visualweather/).

### 5 The impact of the ash cloud on aviation for the Etna eruption scenario

### 5.1 Air Navigation Service Provider

At this demonstration exercise, the tasks designated to ANSPs during such a crisis situation were executed by Austro Control (ACG). ACG provided specially tailored warnings and/or products for different aviation stakeholders in order to warn about the presence of the volcanic ash cloud. During the course of the demonstration exercise, the datasets described in the previous sections were fed into the visualisation software that is also used for daily operations. Subsequently, products were generated

for the local situation in Austria in order to give the exercise participants an impression of available aviation products for the case of a volcanic ash eruption. The presented products included internationally prescribed and harmonized weather warnings or prediction products, as well as country specific products for Austria. In addition, a pilot briefing was conducted including specific information on the volcanic ash event for a potentially affected flight.

The following internationally harmonized products (selected examples) were prepared (according to an ACG internal guideline):

- Volcanic ash SIGMET (Significant Meteorological Information)

  A SIGMET information is a warning message of the occurrence or expected occurrence of specified en-route weather phenomena, which may affect the safety of aircraft operations (ICAO, 2018a). MWO Vienna issued a volcanic ash

SIGMET for April 22, 2018, at 16:00 UTC. It describes that a volcanic ash cloud originated from Mount Etna; the cloud is predicted at 17:00 UTC between FL250 and FL350 in the Austrian airspace within the polygon of which the longitude and latitude coordinates are given (grey line in Fig. 11). Around 18:00 UTC, the ash cloud spread to the southeast and a second polygon is provided (in red in Fig. 11). These polygons were generated from the 75[th] percentile of the ensemble model output (see section 4.3).

- Volcanic ash NOTAM (Notice to Airmen)

  A NOTAM is a notice containing information concerning the establishment, condition or change in any aeronautical facility, service, procedure or hazard, the timely knowledge of which is essential to personnel concerned with flight operations (ICAO, 2018b). The NOTAM, issued for April 22, at 17:58 UTC by MWO or the NOTAM office Vienna, is valid from 18:00 to 21:00 UTC. The predicted volcanic ash of Mount Etna has a high concentration (above 4

20   mg/m3) in the Austrian airspace within the defined polygon.

- Special-Air Reports (AIREP)

  An AIREP is a report of the actual weather conditions as encountered by an aircraft in flight. If pilots observe certain weather phenomena, such as moderate or strong turbulence or icing or the sighting of volcanic ash, a so-called SPECIAL AIREP is issued by the corresponding MWO (International Civil Aviation Organization, 2001) MWO

Rome (represented by ACG) issued this Special AIREP for April 18, at 13:34 UTC. The message contains a pilot report of a volcanic ash sighting at position N3835 E01519 (approximately 60 NM north of Mount Etna) at FL340.

In addition, non-harmonized meteorological products specific for Austria were prepared. In the event of volcanic ash occurring in the Austrian airspace or the directly adjacent airspace, these products were:

- Low-level significant weather chart (Fig. 12): displaying significant weather phenomena below FL250, for the entire Alpine region as well as the adjacent regions.

- Significant Weather Bulletin: an ACG internal weather forecast product for air traffic controllers, a six-hour forecast for weather phenomena that can be significant for en-route traffic and can lead to disruption of the air traffic.

## 5.2 Austrian Air Forces (AAF)

As volcanic ash-clouds are very rare events the respective procedures on the military side are not trained frequently. The EUNADICS-AV demonstration exercise was a good opportunity for the Austrian Airforce to test the following tasks:

- Prove the AAF-concept for volcanic incidents in a European civil-military context.
- Represent possible information needs of European Airforces
- Demonstrate, that each Airforce will have its own national specific tasks (e.g. Austria has to ensure neutrality and sovereignty 24/7).
- Explain military activities that might occur during real volcanic ash events.
- Present the AAF „air sampling" capability in the European civil-military context and highlight its value for European forecast and dispersion modelers.

To fulfil above mentioned tasks, AAF participated with the following key personnel of the Austrian Air Operations Centre (AOC): Chief Current Operations, Air Space Manager, chemical, biological, radiological and nuclear (CBRN) specialist, two military meteorologists and a military programmer.

The AAF made an internal reprocessing of the real air operation scenario "Informal meeting of justice and home affairs ministers 2018 in Salzburg" under the influence of volcanic ash-clouds with relevant ash concentrations in Europa and over Austria. The AAF demonstrated how airspace blockings in the southern part of Europe might influence military overflights through Austrian airspace.

## 5.3 Rerouting of flights

One of the exercise objectives was to demonstrate how efficient, automated future airline operations in a disaster scenario could function. Impacts on ATM were shown for both, the civil and the military sectors within each phase of the exercise. For evaluating the impacts on the air traffic, the air traffic was simulated by using the NAVSIM/USBGSim simulator.

NAVSIM/USBGSim is an ATM/ATC/CNS (Communication, Navigation & Surveillance)/MET (meteorology) simulation framework developed and continuously enhanced by the Aerospace Research Department from the University of Salzburg, in close cooperation with Mobile Communications R&D (Rokitansky et al. 2007a and 2007b, Rokitansky et al. 2018a and 2018b).

NAVSIM/USBGSim has been used to simulate European and world-wide air traffic based on specific reference days in the past (around 36000 flights within 24 hours for Europe and 110000 flights worldwide). It can be used as real-time and fast-time simulator.

A total of around 243000 flights between April 18 and April 25 of year the 2018 were analysed and simulated for the demonstration exercise using post-flight input data from the Eurocontrol/Network Manager (NM), which means around 30400 flights per 24 hours. In principle, all scheduled airline, charter and cargo flights, as well as short term business, general aviation

and military flights, which entered a controlled air space within the ECAC[1] area were taken into account. The detailed flight route of each of these "real" flights was used as basis for further comparison with the volcanic ash scenario assumed and modelled for the EUNADICS-AV exercise. A total number of around 98000 flights were identified for potential conflicts with ash clouds. For each of these intersecting flights suitable deviation routes – if required – were calculated by the EUNADICS-

AV participant FlightKeys and further simulated and visualized using the NAVSIM/USBGSim tool of University of Salzburg. Selected flights were simulated in real time using the Laminar Research's flight simulator (X-Plane 11, 2019) and were visualized on-line in NAVSIM/USBGSim. Furthermore, voice communications concerning visually observed volcanic ash were exchanged in real time between involved "humans in the loop" respectively a pilot and an air traffic controller for selected flights.

A newly developed cost model was used in the framework of an airline network balance tool (Flightkeys 5D, https://www.airlinesoftware.net/product/1422/flightkeys-5d) to cost-efficiently re-route flights affected by the disaster event. A realistic damage cost model has been developed with the advice of the aircraft engine manufacturer Rolls-Royce, and the Flightkeys trajectory optimizer was modified to integrate the cost model and 5D[2] ash data. ATM flow restrictions and mandatory routes were disabled to reduce complexity and to allow more efficient re-routings. Since the vast majority of flights

were unaffected and could be assumed to proceed according to their originally filed routings, network disturbance was assumed to be manageable at a level that would be similar to a large convective weather situation. Furthermore, the assumption was made that a future Europe-wide ATM system would be capable to accommodate the rerouting requests in an efficient way for the case where practically all airspace users would utilize the advanced trajectory planning capability and all have Safety Risk Assessments that allows flight through/over ask clouds.

The "domino" effect of delayed or cancelled flights on connecting flights was not simulated in the exercise. A guess on that effect would be a doubling of the predicted flights cancellation rate.

The Flightkeys system during the EUNADICS-AV exercise is shown in Fig. 13. Hi-resolution volcanic ash data for the entire scenario period (April 18 to 25, 2019) was imported as 75 percentile polygons (83000 polygons) with a temporal resolution of 1 hour. Polygons were imported for the following intervals: <1, 1=<2, 2=<3, 3=<4 and >=4 mg/m3. A large-scale, flight by

flight, optimization was performed on the entire set of 98000 flights, re-optimizing them vertically and laterally, considering the following factors:

1.   Upper air wind and temperature (GRIB data in 6h time resolution, 1.25° lateral and 2000ft vertical)
2.   Hi-resolution ash model data (see above)

---

[1] European Civil Aviation Conference (ECAC) currently includes 44 member states (refer to https://www.ecac-ceac.org/member-states).

[2] In addition to the 4D space covered by traditional flight planning solutions, 5D extends the calculation space into a 5th dimension. Uncertainties in surface weather, traffic and cost prediction is modelled into statistical functions based on a continuous analysis of actual flight data. For upper air weather, multicasting weather products are introduced to compare multiple scenarios and automatically apply suitable strategies, e.g. adaptive fuel reserves and delay cost reduction (for more information see www.flightkeys.com/).

3. Detailed aircraft performance (not the BADA model, but OEM-provided flight planning data)

4. Cost of reduced engine lifetime due to ash damage

5. Cost of increase in future fuel consumption due to ash damage and accumulation

6. Effect of air density on ash mass ratio

7. Effect of fuel flow on ash accumulation in engines (high fuel flow = high air flow = more ash accumulation)

It was demonstrated that the flight trajectories successfully avoided the ash cloud (see examples in Fig. 14 & Fig. 15) in the most economical way by applying a newly developed algorithm that predicts future maintenance cost and fuel efficiency losses, even in complex ash-cloud / airway situations (airways only provide a very limited set of possible flight paths).

The algorithm is based on the assumptions that air flow through the engine core is proportional to fuel flow and thus allows a direct correlation of ash accumulation to fuel flow. Rolls-Royce contributed first estimated ratios of ash mass accumulation versus engine deterioration (e.g. exhaustgas-temperature margin loss per kg of ash), thus allowing to predict both engine efficiency and engine lifetime decreases and their cost equivalents. The direct relationship to fuel flow is extremely well suited to the already well-established cost optimization algorithm of FLIGHTKEYS' flight planning system.

With full availability of free-route airspaces across the entire ECAC airspace, even better avoidance trajectories can be expected. Since maintenance cost rises sharply already at low ash concentrations, the ash cloud was often avoided entirely.

Analysis of the large-scale optimization run showed that 2% of the considered flights had to be considered as "hard" cancellation candidates as a direct consequence of excessive ash concentrations along their routes, while 7% of the analysed flights required 4D-tailored trajectories. As mentioned above, an estimate of cancellations due to aircraft or crew rotation

aspects would probably lead to a doubling of that cancellation rate to 4% which is still very low. One important conclusion was that the majority of flights were practically unaffected.

**6 Conclusion**

During the EUNADICS-AV exercise, it was demonstrated that tailored and selected observations as well as dedicated model applications can successfully support aviation stakeholders in their decisions during an aviation crisis situation such as a

dispersing volcanic ash cloud. A key objective of the exercise was to include a cost-benefit approach within the decision-making process.

One major result has been a presentation of the benefit of using a single harmonised portal for comprehensive visualisation of data and products. Such a platform enables crucial timesaving during a crisis, when decisions must be made fast and efficiently. Once the data is visualised and processed, the possibility of using it integrated into flight modelling software enables a more

effective decision-making process with the re-routing done in an automatic or quasi-automatic approach. For our particular case, despite the relevant event simulated and its proximity to the highly busy airspace, the exercise resulted in the surprising and useful result that by integrating ash contamination effects into cost-based trajectory optimization algorithms, most of the flights are almost unaffected ~~and remaining airspace is much better utilized than~~ by crude manual avoidance methods. The

estimated impact was therefore much lower than initially expected.  For volcanic events, a cost-benefit approach was adopted, whereby only flights above a certain threshold would be cancelled. The cost-benefit calculations revealed only for a small number of cases that cancellation was an economically better option than the execution of the flight. The cancellation threshold was set at additional cost exceeding 200% of all other considered operating costs (fuel, time, overflight charges).

In the end, every air space closure or even re-routing of planes can immediately increase the costs for airlines which will lead to a certain risk acceptance, at least through regions which lie in an area with the ash concentration above the threshold. During the exercise we have shown for a volcanic ash scenario, that cost and disruption ~~of air traffic~~ can be eliminated to a great extent by combining dispersion models with flight planning software to apply cost-based trajectory optimizations developed within the EUNADICS-AV project.

It can be disputed how big such positive effects would be in the current, fragmented and over-regulated European airspace, but this only underlines how important future progress in the automation and unification of ATM systems and processes will be to allow more flexibility in airspace disruption scenarios like the ones simulated in the exercise

Such conclusions would not be possible if the working practices and interfaces between the broader natural hazard science communities on one side and the more application-oriented aviation community on the other side, mostly represented in the

project by flight trajectory modelers and military aviation practitioners, would not have been established. The intense preparatory work ahead of the exercise managed to bridge existing gaps and bring together experts that have not cooperated that closely before. This exercise, and the EUNADICS-AV project as a whole, has provided the very first steps towards integrating an impact-oriented perspective.

**7 Acknowledgements**

This work has been conducted within the framework of the EUNADICS-AV project, which has received funding from the European Union's Horizon 2020 research programme for Societal challenges - smart, green and integrated transport under grant agreement no. 723986. We thank the Austrian military for hosting the exercise and providing technical equipment together with the University of Salzburg.

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

**Glossary**

|  |  |  |
|---|---|---|
| AAF | - Austrian Armed Forces |
| ACG | - Austro Control |
| ACTRIS | - Aerosols, Clouds, and Trace gases Research InfraStructure |
| AFTM | - Aeronautical Information Services and Air Traffic Flow Management |
| AIREP | - Special-Air Reports |
| ANSP | - Air Navigation Service Providers |
| AOC | - Air Operations Centre |
| AOCCC | - Aircraft Operator Crisis Coordination Cell |
| AOD | - Aerosol Optical Depth |
| a.s.l | - above sea level |
| ATC | - Air Traffic Control |
| ATM | - Air Traffic Management |
| BADA | - Base of Aircraft DAta |
| CAAs | - Civil Aviation Authorities |
| CBRN | - Chemical, Biological, Radiological and Nuclear |
| CNS | - Communication, Navigation & Surveillance |
| ECMWF | - European Centre for Medium Range Forecasts |
| EACCC | - European Aviation Crisis Coordination Cell |
| EARLINET | - European Aerosol Research Lidar Network |
| ECAC | - European Civil Aviation Conference |
| EUNADICS-AV | - European Natural Disaster Coordination and Information System for Aviation |
| Eurocontrol | - European Organisation for the Safety of Air Navigation |
| ERA-5 | - ECMWF Re-Analysis |
| EWS | - Early Warning System |
| FL | - Flight Level |
| FLEXPART | - FLEXible PARTicle dispersion model |
| GRIB | - GRIdded Binary |
| IASI | - Infrared Atmospheric Sounding Interferometer |
| IAVW | - International Airways Volcano Watch |
| ICAO | - International Civil Aviation Organisation |
| INGV | - Italian National Institute of Geophysics and Volcanology |
| KPI | - Key Performance Indicator |

| | | |
|---|---|---|
| LIDAR | - Light Detection And Ranging | |
| MATCH | - Multi-scale Atmospheric Transport and Chemistry | |
| MetOp | - Meteorological Operational Satellite | |
| MOCAGE | - Modèle de Chimie Atmosphérique de Grande Echelle | |
| MODIS | - Moderate Resolution Imaging Spectroradiometer | |
| MWO | - Meteorological Watch Office | |
| NAVSIM/ USBGSim | - Navigation Simulator (University of Salzburg) | |
| NM | -Network Manager | |
| NOTAM | - Notice to Airmen | |
| NRT | - Near Real Time | |
| OEM | - Original Equipment Manufacturer | |
| PIREP | - Pilot Report | |
| PMA | - Parts Manufacturer Approval | |
| SCC | - Single Calculus Chain | |
| SIGMETS | - Significant Meteorological Information | |
| SILAM | - System for Integrated modeLling of Atmospheric coMposition | |
| STC | - Supplemental Type Certificate | |
| VAACs | - Volcanic Ash Advisory Centres | |
| VACP | - Volcanic Ash Contingency Plan | |
| VOLCEX | - VOlcanic Ash Contingency EXercise | |
| VONA | - Volcanic Observatory Notice for Aviation | |
| WMO | - World Meteorological Organization | |
| WRF-Chem | - Weather Research and Forecasting (WRF) model coupled with Chemistry | |

**Figures**

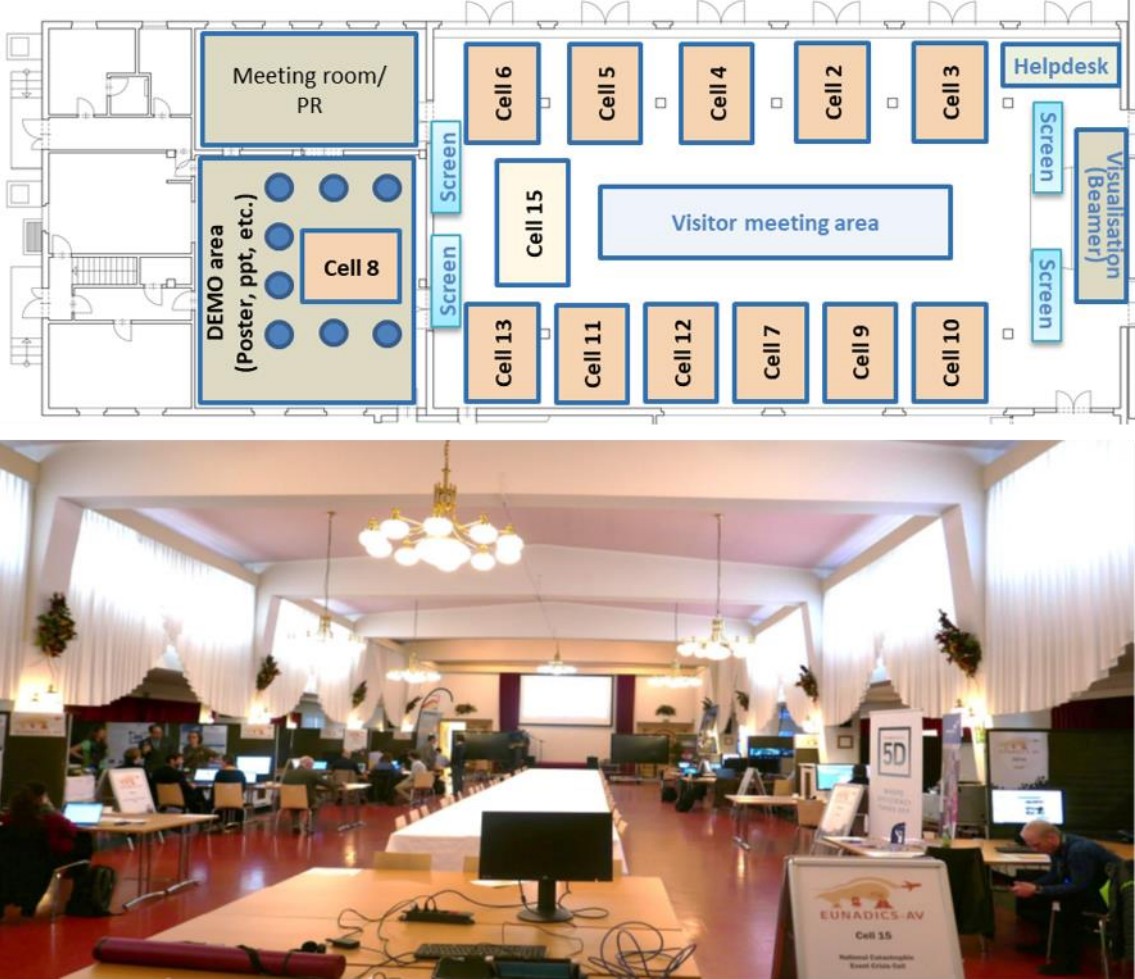

**Figure 1: Top: floor plan of different cells at Schwarzenberg barracks (Salzburg, Austria). Bottom: Picture of the main hall of the**
**demonstration exercise premises.**

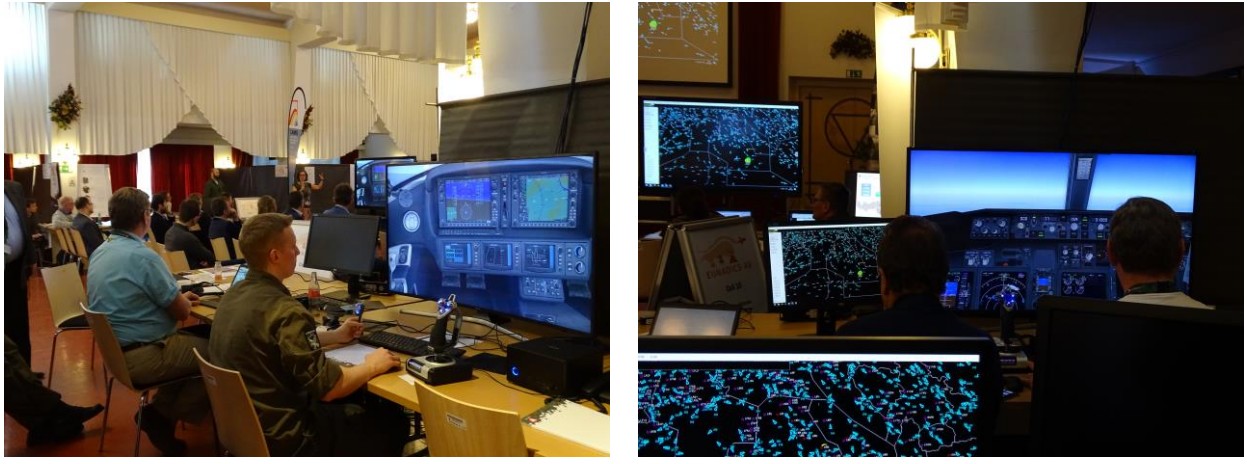

**Figure 2: Selected photos from the demonstration exercise at the Schwarzenberg barracks in Salzburg. The flight simulator was operated by both military and civil pilots.**

# EUNADICS-AV Exercise

|  | (1) | (2) | (3) | (4) | (5) |
|---|---|---|---|---|---|
| Natural hazards affecting aviation | Data acquisition | Early warning | 4D model analysis | Model ensemble | Data to aviation user |

- **Exercise 1:**
  - Volcano
- **Exercise 2:**
  - Nuclear event
- **Demonstration examples:**
  - Volcanoes
  - (Forest) fires
  - Dust storms
  - etc.

Available satellite data

In-situ measurement data

Quick information to (end) users

Central data platform

Model 1

Model 2

Model 3

...

Ensemble

Central hazard event database

Event

1 2 3 4

Pre-alert  Disruption  Crisis  Recovery

5    **Figure 3: EUNADICS-AV exercise 2019 workflow.**

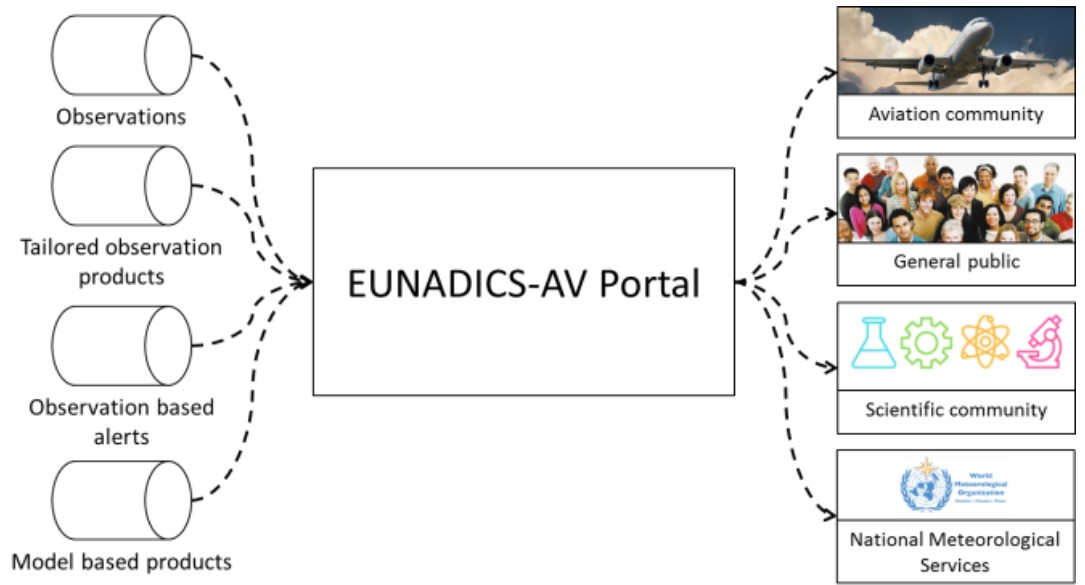

**Figure 4: Information flow between data providers and users.**

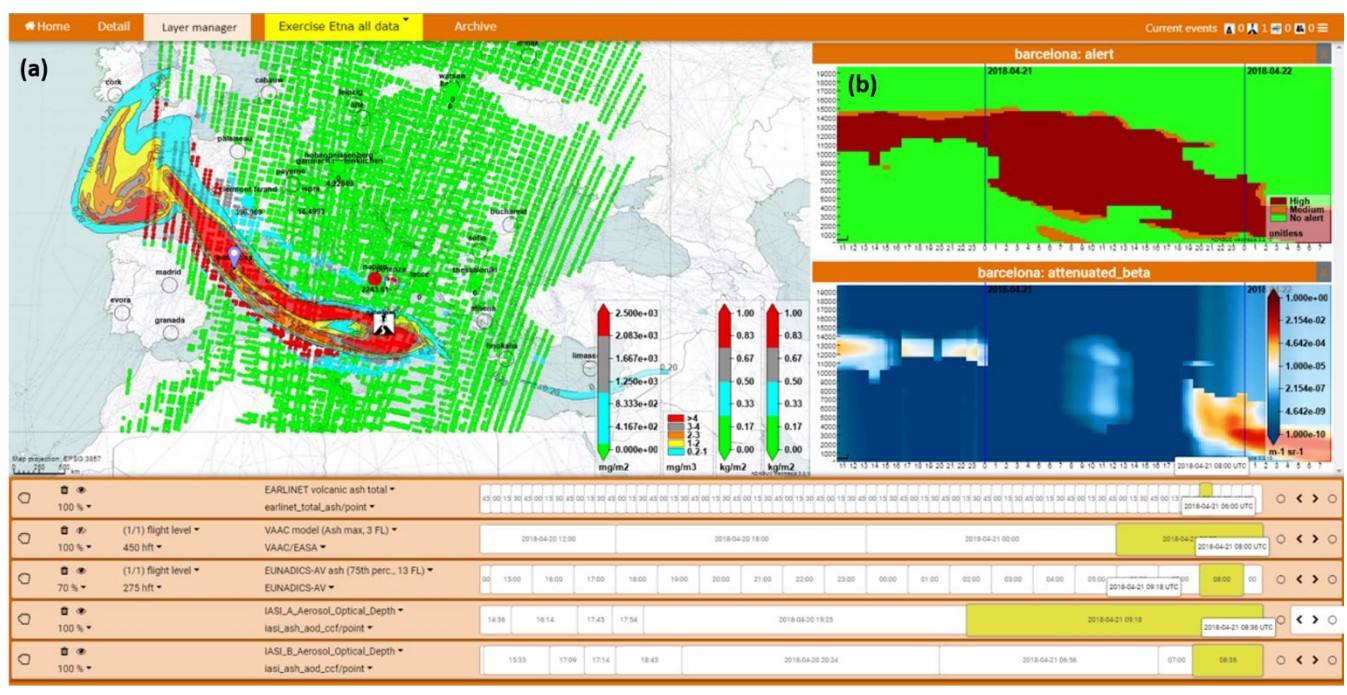

**Figure 5: Exemplified visualization of the EUNADICS-AV portal showing the model ensemble (a) and the vertical ash distribution (b) at an EARLINET/ACTRIS station (Barcelona) on April 21, 2018.**

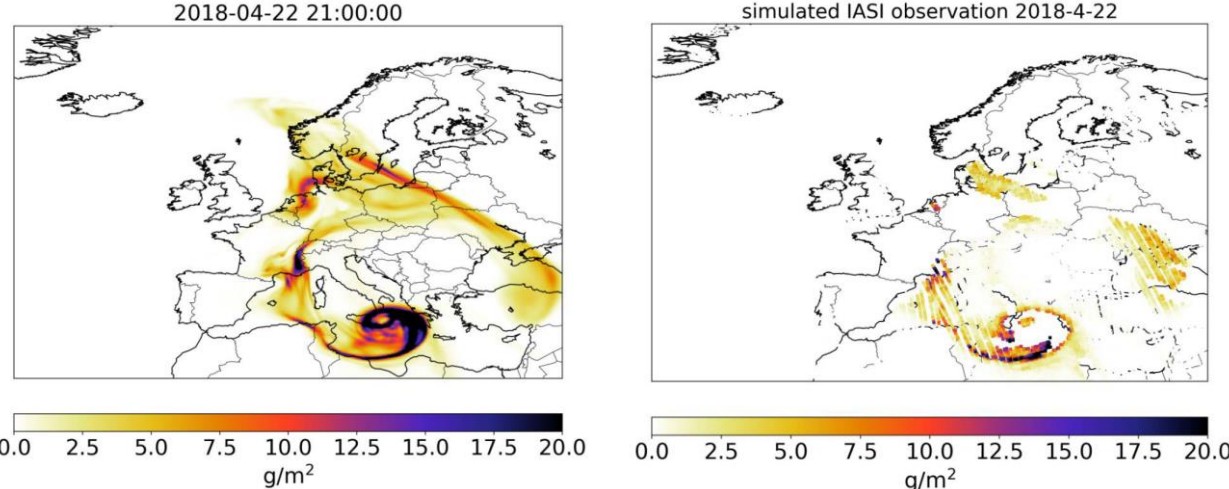

**Figure 6: Left: Simulated observations based on actual IASI pixels; no retrievals if: cloud cover fraction > 0.25 in the 0.1° x 0.1° model pixel. Right: simulated AOD > 8. The values are perturbed with a spatially correlated error (the spatial correlation is based on a similar error covariance as the simulated MODIS AOD)-the error of each data point is set to 0.25 times the simulated value plus 0.05 g m-2.**

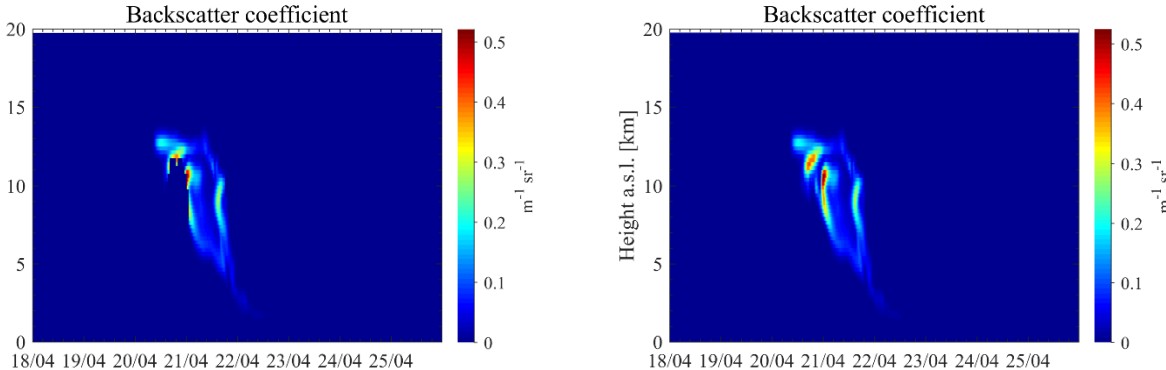

**Figure 7: Particle backscatter coefficient time-height evolution when the cloud fraction is removed (left panel) and when not (right panel) for the Barcelona EARLINET station and for the whole exercise period.**

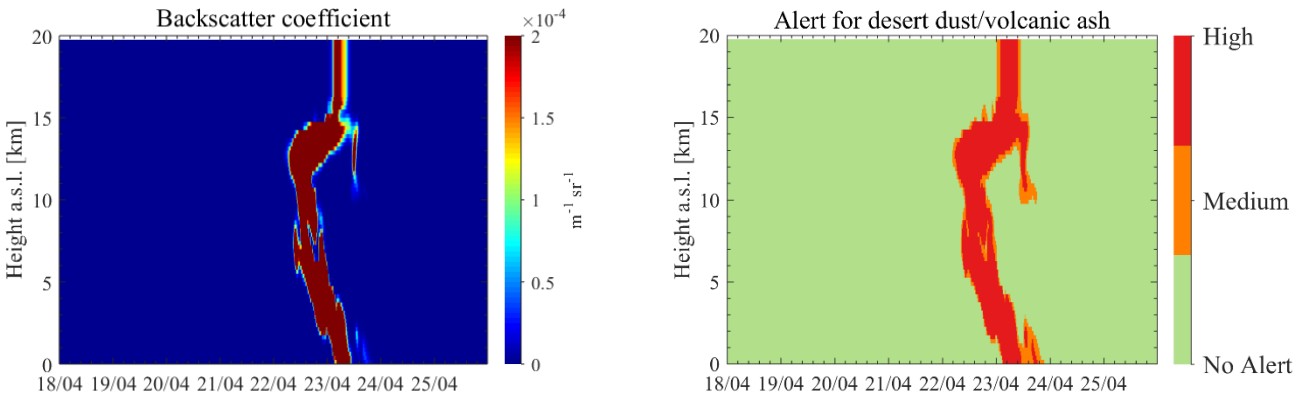

**Figure 8: The attenuated backscatter coefficient (left panel) and the alert for aviation (right panel) for the Leipzig EARLINET station.**

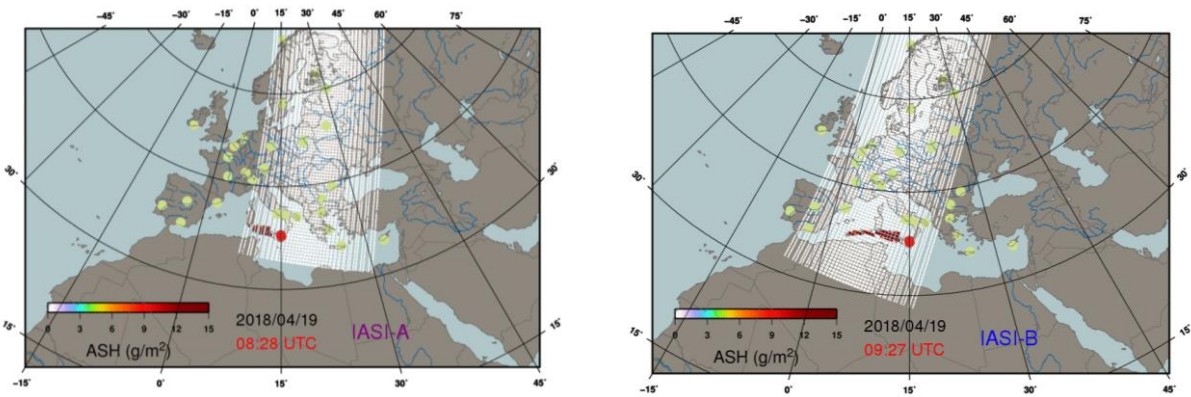

**Figure 9: Warnings of ash plume by IASI-A (at 08:28 UTC) and IASI-B (at 09:27 UTC) on April 19, 2018.**

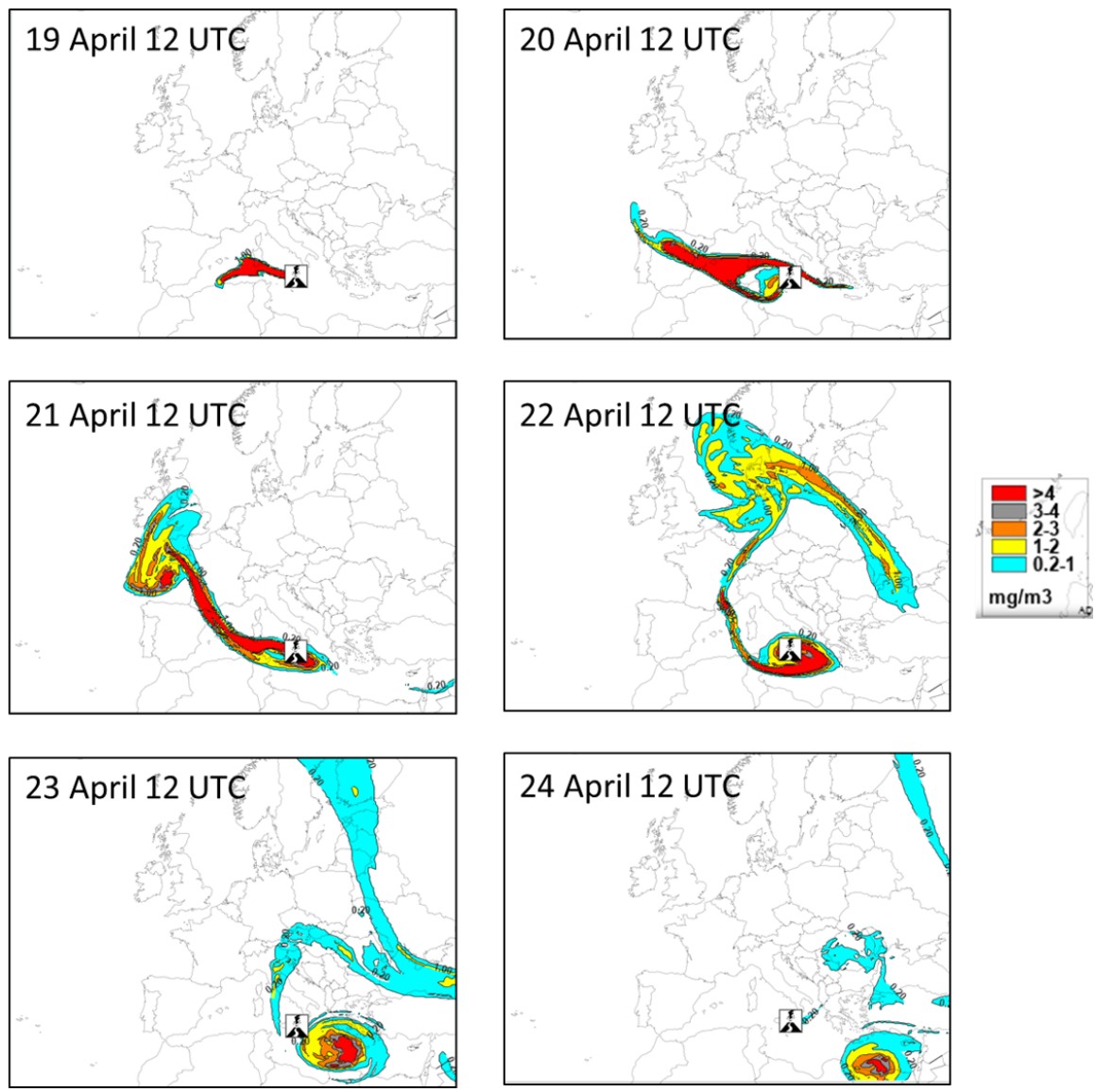

**Figure 10: Model ensemble (75-percentile) for the 6 consecutive days after the artificial Etna eruption in April 2018. The data is depicted on FL275.**

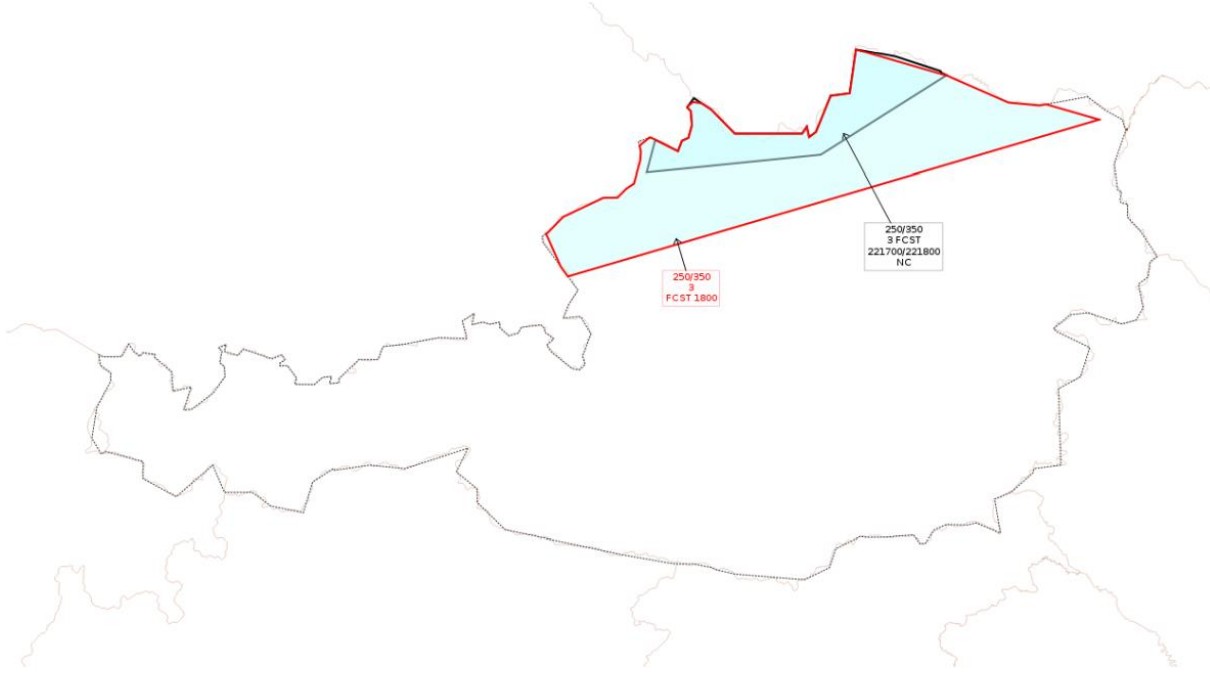

**Figure 11: Location of the two polygons for which the SIGMETS (ACG) were issued.**

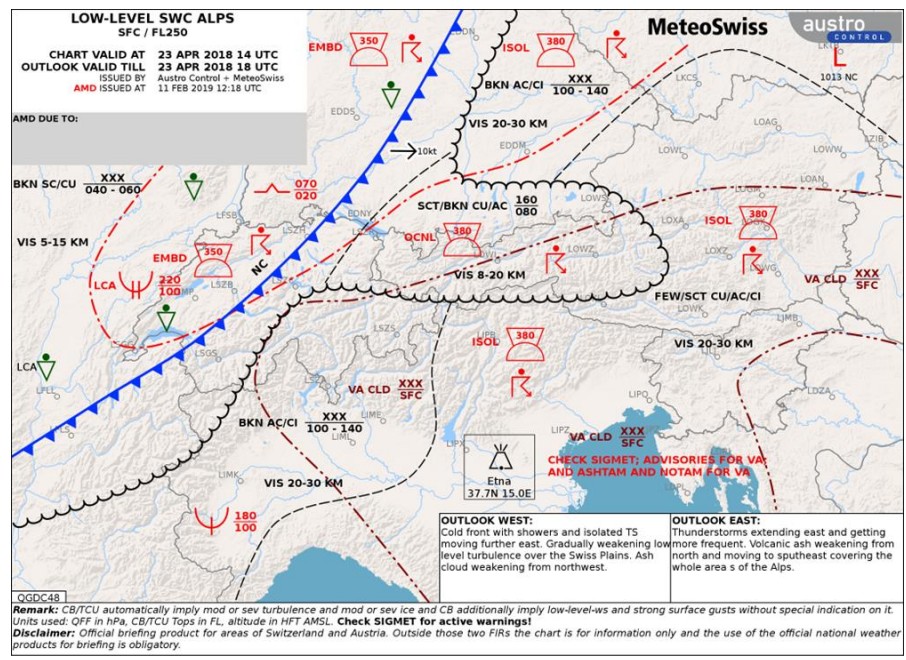

**Figure 12: Low-level significant weather chart for April 23, 14:00 UTC. The predicted volcanic ash is represented by the brown**
5   **dash-dot-dotted line, over the Adriatic Sea, parts of Italy and Slovenia, the south-eastern Switzerland and the southern half of Austria and Hungary extends from the surface to over FL250.**

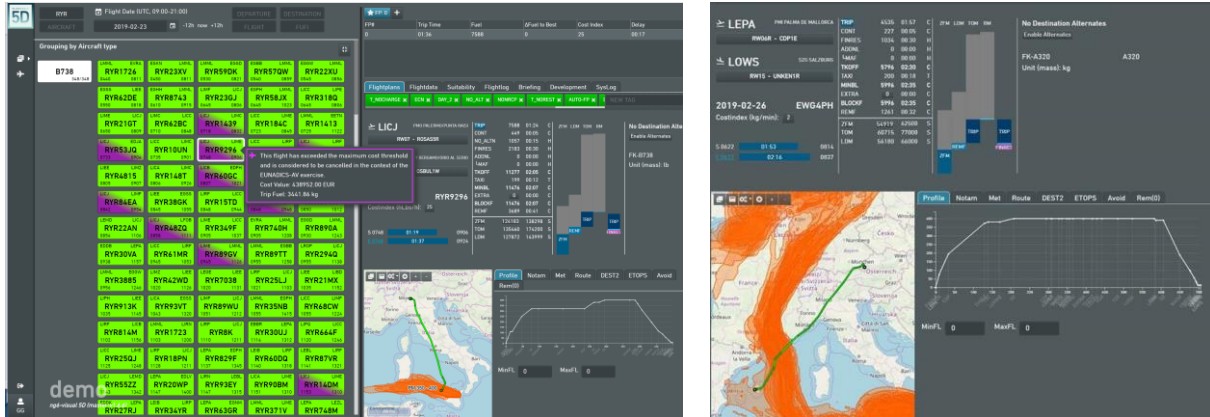

**Figure 13: Flightkeys system during the EUNADICS-AV exercise. Left: overview of all monitored flights. Right: status (location, height, planned route) of an individual flight.**

**Figure 14: Volcanic ash (red: >=4 mg/m3, yellow, 2-4 mg/m3, green 0.2-2 mg/m3) reroute calculated by FlightKeys and simulated with NAVSIM/ USBGSim (University of Salzburg), laterally and vertically avoiding ash concentration.**

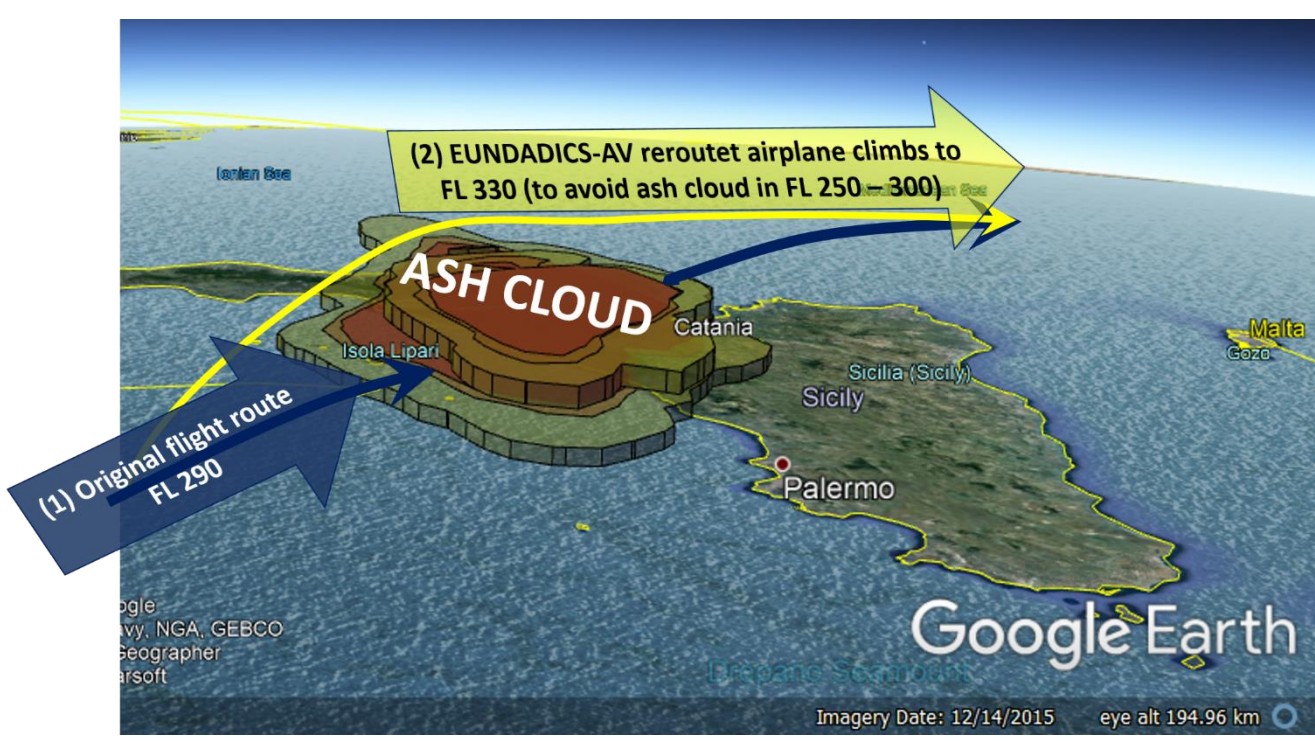

Figure 15: Example of a rerouting of one flight calculated by FlightKeys and simulated with NAVSIM/USBGSim (University of Salzburg), avoiding laterally and vertically vulcanic ash (red: >=4 mg/m3, yellow, 2-4 mg/m3, green 0.2-2 mg/m3 in different FLs).

**Tables**

**Table 1: EUNADICS-AV demonstration exercise cells and related tasks.**

| No. | Cell name | Tasks |
|---|---|---|
| C 1 | EUNADICS-AV Exercise Lead | Management team of EUNADICS-AV demonstration exercise (was located in an extra room, not depicted here). |
| C 2 | Observational Infrastructure | Collection, tailoring, documentation, and distribution of observational data such as satellite and lidar data. |
| C 3 | Early Warning System | Sending of early warning messages and triggering of the EUNADICS-AV modelling chain based on the observational data provided by C 2. |
| C 4 | Data integration and assimilation | Based on observations, estimates of the source term (location and strength) will be provided. Furthermore, analysis charts are computed by assimilating relevant observations into numerical models. |
| C 5 | Aviation product development and integration | Collects the model data from the individual modelling groups, which are produced in C 4, and computes a model ensemble which is passed further to the EUNADICS-AV portal where further output concentration charts and processing for the Air Traffic Management (ATM) is conducted. |
| C 6 | Data and product delivery | Distribution of relevant products and visualization of different data sets in a harmonized way. Interface between EUNADICS-AV products and end users. |
| C 7 | Aviation specific data usage and analysis | Aviation specific use of EUNADICS-AV products, simulation of air traffic (civil/military perspective), flight management, etc.. |
| C 8 | User, risk assessment | Visitor coordination, user survey during the event and collection of feedbacks and additional requirements. |
| C 9 | Pilots | Demonstration of a pilot's view. |
| C 10 | Military/air forces | Demonstration of military application of EUNADICS-AV data and products. |
| C 11 | Airports | Demonstration of an airport's view in case of a disaster event. |
| C 12 | Airlines | Demonstration of cost-based decision making in case of a disaster event. |
| C 13 | Air Navigation Service Providers (ANSPs) | Role and application of ANSPs in case of a disaster event. |
| C 14 | EUNADICS-AV Crisis Coordination Cell | Coordination of the exercise crisis cell and exercise-specific events. |

| C 15 | **National Catastrophic Event Crisis Cell** | Role and applications of national authorities in case of a disaster event. |
|------|------|------|