# Peer review of "A volcanic hazard demonstration exercise to assess and mitigate the impacts of volcanic ash clouds on civil and military aviation"

_Natural Hazards and Earth System Sciences, 2019_

## Referee Comment (RC1) · Tatjana Bolic (Referee) · 14 Nov 2019

Important re-framing of the section 5 and consequently section 6 (conclusions) is needed, as the assumptions taken in the flight optimizations and simulations prevent from reaching the conclusions the authors wrote.

The authors state they do not take into account the airway structure (that is OK, as in crisis situations these are also removed), and the ATM flow restrictions. If the ATM flow restrictions are not taken into account then we can show that any intervention will bring marvelous results. However the ATM restrictions exist in the real life, and are there for the safety reasons - they do complicate and constrain the overall traffic, it is

true. Other important things not taken into account (as are not mentioned, so I assume are not taken into account) are the individual airline Safety Risk Assessment - that is to say what decisions they take when ash is forecasted - a significant percentage of airlines would not fly in any forecasted concentration of ash. Further, the connections between flights are not taken into account - if an aircraft that is supposed to perform the flight does not arrive (or does not arrive in time), then the next flight will not happen (or will be delayed). It also does not say anything if the simulator keeps the aircraft separated (I assume it does, but it is not mentioned). As these things are not taken into account, the conclusion that only a small percentage would be cancelled and that most of the flights would not be affected are not valid! Take into account the presence of the weather fronts (which are a bit less dangerous to aircraft, and on which we have better information) creates havoc in the network - cancellations, enormous delays, then cancellations due to extensive delays, etc.

What probably should be better stressed is the inclusion of the maintenance cost in the trajectory optimisation which has an important impact on managing to automatically adjust the flight trajectory away from the cloud. As that is I think the most important point here. What is really important that came out of this exercise is the fact that if the maintenance costs are used in the flight trajectory planning software that optimises on costs, then the airlines can easily obtain trajectories that avoid the highest (and lower?) concentrations of the ash. At the moment, most of the trajectory planning in the presence of ash cloud has to be done manually or by manually imposing airspace restrictions in the trajectory planning tools. And are not taking into account the maintenance costs.

There are several other, smaller requests for clarification: 1. The difference between this demonstration exercise and actual operational setting in the ATM, should be mentioned, and you should cite the ICAO volcanic ash procedures for the EU region https://www.icao.int/EURNAT/EUR%20and%20NAT%20Documents/EUR+NAT%20VACP.pdf 2. Need to specify why certain concentration level thresholds were taken. 3. KPIs are

mentioned several times in the paper, but are not elaborated on. Either elaborate - show which ones and how they were used, or do not mention them.

Please also note the supplement to this comment:
https://www.nat-hazards-earth-syst-sci-discuss.net/nhess-2019-265/nhess-2019-265-RC1-supplement.pdf

---

## Referee Comment (RC2) · Andreas Vogel (Referee) · 9 Dec 2019

General comments: The presented approach of flight optimisations and simulations during natural airborne aviation hazards is a novel and important concept to minimise disruptions during a volcanic crises. However, the submitted manuscript has some shortfalls that need addressing. The described exercise was technically challenging and apparently a lot of planning went into the different modelling and calculation aspects. Unfortunately, the manuscript does not fully reflect this. In order to improve the manuscript and to emphasise the importance of such an approach some of the manuscript sections require reworking. Furthermore, the different aspects are not out-

lined clearly which makes it hard to follow to understand the individual details (clarity of concept). It can be assumed that the target audience of the manuscript will be geologists, atmospheric and earth sciences scientists as well as the flight planners, decision makers and engineers. Therefore, it is recommended to explain some aspects in section 3 in more detail. An overview chart (more detailed than Fig. 3) would help guide the audience through the manuscript. Section 5 and 6 need most of the reworking. The drawn conclusions do not reflect the archived aspects and novel concept. Another aspect the manuscript forfeits is to describe the difference between the current flight planning approach and the novel approach. That includes model assumptions, safety risk assumptions as well as airway structure and ATM flow restrictions. Only if the difference is clearly described and compared, the reader can understand the benefits of such an approach. The novel concept i.e. use of maintenance costs are used in flight planning software were mentioned but a more detailed description of those methods would be useful to better understand and stress the new approach. In summary, the submitted manuscript can be a valuable contribution to future flight planning and tasks, but in the current state does not exhaust its potential. A substantial reworking of the manuscript can be beneficial and is strongly suggested.

Addressing individual scientific questions/issue:

1. The difference between the presented flight planning approach (described exercise) and the current operational approach is not clear or sufficiently described. Also, the link to ICAO and airline procedures could be further elaborated.

2. Describe concentration levels and why certain levels are acceptable and certain level not with the main focus on aircraft engines as the main critical parts.

3. The text includes a lot of names/acronyms that are sometimes only mentioned once. Perhaps it would be better to not introduce acronyms if the name is only mentioned once. This would allow a better readability.

4. Describe the mentioned factors in section 5.2 (page 12) further. This is needed to

better understand the novel approach.

5. Statement in conclusion about "future natural disasters" can be misleading. The exercise was setup for volcanic ash particulates. The situation, for example sulphur dioxide, might be completely different, e.g. different detection, distribution, aircraft and engine threats etc.

6. Figures and Table are a bit detached from the text. It would be welcomed to add some more "scientific" plots to explain the procedure a bit more. Perhaps in combination with an overview chart. Table 1 is not even mentioned in the text.

Please also see the supplement to this comment.

Please also note the supplement to this comment:
https://www.nat-hazards-earth-syst-sci-discuss.net/nhess-2019-265/nhess-2019-265-RC2-supplement.pdf

**Supplement:**

[Figure]

[Figure]

[revised manuscript text omitted]
 of the cells provided additional information on the used data sets and model applications as well as the scientific background of the procedures. Each cell was equipped with technical support, e.g., computers and monitors to demonstrate its role in the operational sequence of procedures and actions. Main results could be projected from each cell to various big

20    screens in the hall (see Fig. 2).

Based on the data provided (see section 4) by the "scientific cells" (cell 2 to 6), the impact on aviation was simulated and depicted by the "aviation and stakeholder cells" (cell 7 to 15). These data allowed the ANSPs to release special aviation advisories (e.g. Significant Meteorological Information - SIGMETS). The impact of the event was recorded by keeping track of all aspects related to economically guided decision making in airline operational centres. Further impacts were due to re-

25    routing and cancellation of flights by flight trajectory modellers (see section 5.3) as well as respective procedures invoked on the military side and participating airports and ministries. For the evaluation of the impacts on the air traffic, the air traffic was simulated by using the NAVSIM (Rokitansky, 2009) simulator and was analysed by using Key Performance Indicators (KPIs). A newly developed cost model was used in the framework of an airline network balance tool (Flightkeys 5D) to cost-efficiently re-route flights affected by the disaster event (see section 5.3). Cell-specific EUNADICS-AV developments and

30    impacts on ATM were shown for both, the civil and the military roles within each ATM phase.

The most important tasks in the preparation phase of the demonstration exercise, were to establish working practices and interfaces between the broader natural hazard science communities on one side and the more application-oriented aviation community on the other side. The latter was mostly represented by flight trajectory modellers and military aviation experts.

[Figure]

The intense preparatory work ahead of the demonstration exercise managed to bridge existing gaps. Experts such as natural hazard scientists to flight managers and pilots collaborated in an unprecedented fashion. This effectively was the spirit of the EUNADICS-AV project put into practice.

**3.1.1 The volcanic eruption scenario**

5  The exercise scenario was designed assuming an eruption of Etna volcano in Sicily, Italy. The period of the eruption was selected to guarantee an atmospheric circulation situation that favoured the transport of ash over large parts of Europe. 
[revised manuscript text omitted]
 (see https://www.skybrary.aero/index.php/Pilot_Report (PIREP)). MWO Rome issued this Special AIREP for April 18,

at 13:34 UTC. The message contains a pilot report of a volcanic ash sighting at position N3835 E01519 (approximately 60 NM north of Mount Etna) at FL340.

In addition, non-harmonized meteorological products specific for Austria were prepared. In the event of volcanic ash occurring in the Austrian airspace or the directly adjacent airspace, these products were:

- Low-level significant weather chart (Fig. 11): displaying significant weather phenomena below FL250, for the entire Alpine region as well as the adjacent regions.
- Significant Weather Bulletin: an ACG internal weather forecast product for air traffic controllers, a six-hour forecast for weather phenomena that can be significant for en-route traffic and can lead to disruption of the air traffic.

**5.2 Austrian Air Forces**

As volcanic ash-clouds are very rare events and the respective procedures on the military side are not trained frequently. The EUNADICS-AV demonstration exercise was a good opportunity for the Austrian Airforce to test the following tasks:

- Prove the Austrian AAF-concept for volcanic incidents in a European civil-military context.
- Represent possible information needs of European Airforces
- Demonstrate, that each Airforce will have their own national specific tasks (e.g. Austria has to ensure neutrality and sovereignty 24/7).
- Explain military activities that might occur during real volcanic ash events.
- Present the AAF „air sampling" capability in the European civil-military context and highlight its value for European forecast and dispersion modelers.

To fulfil above mentioned tasks, AAF participated with the following key personnel of the Austrian Air Operations Centre (AOC): Chief Current Operations, Air Space Manager, Chemical, biological, radiological and nuclear (CBRN) specialist, two military meteorologists and a military programmer.

The AAF made an internal reprocessing of the real air operation scenario "Informal meeting of justice and home affairs ministers 2018 in Salzburg" under the influence of volcanic ash-clouds with relevant ash concentrations in Europa and over Austria. The AAF demonstrated how airspace blockings in the southern part of Europe might influence military overflights through Austrian airspace.

**5.3 Rerouting of flights**

One of the exercise objectives was to demonstrate how efficient, automated future airline operations in a disaster scenario could function. Impacts on ATM (e.g. Fig. 12) were shown for both, the civil and the military sectors within each phase of the exercise. For evaluating the impacts on the air traffic, the air traffic was simulated by using the NAVSIM simulator

developed by the Aerospace Research Department from the University of Salzburg, and analysed by using Key Performance Indicators (KPIs).

NAVSIM is an ATM/ATC/CNS (Communication, Navigation & Surveillance)/MET (meteorology) simulation framework developed and continuously enhanced by the University of Salzburg, in close cooperation with Mobile Communications

5    R&D (Rokitansky et al. 2007a and 2007b, Rokitansky et al. 2018a and 2018b). NAVSIM has been used to simulate European and world-wide air traffic based on specific reference days in the past (around 36000 flights within 24 hours for Europe and 110000 flights worldwide). It can be used as real-time and fast-time simulator.

Around 243000 flights were analysed and simulated for the demonstration exercise using input data from Eurocontrol. A number of 98000 flights were identified for potential intersection with ash clouds. These intersecting flights were used for

10    further calculation of suitable deviation routes. The re-calculated routes were simulated and visualized by NAVSIM (3 seconds time step). Selected flights were simulated in real time using the Laminar Research's flight simulator (X-Plane 11, 2019) and were visualized on-line in NAVSIM. Furthermore, voice communications concerning visually observed volcanic ash were exchanged in real time between involved "humans in the loop" respectively a pilot and an air traffic controller.

A newly developed cost model was used in the framework of an airline network balance tool (Flightkeys 5D,

15    https://www.airlinesoftware.net/product/1422/flightkeys-5d) to cost-efficiently re-route flights affected by the disaster event. A realistic damage cost model has been developed in close cooperation with the Rolls-Royce aircraft engine manufacturer, and the 5D trajectory optimizer was modified to integrate the cost model and 5D ash data. ATM flow restrictions and mandatory routes were disabled to reduce complexity and to allow more efficient re-routings. Hi-resolution volcanic ash data for the entire scenario period (April 18 to 25, 2019) was imported as 75 percentile polygons (83000 polygons) with a

20    temporal resolution of 1 hour. Polygons were imported for the following contour levels: 1.0, 2.0, 3.0, >=4.0 mg/m3. A large-scale optimization was performed on the entire set of 98000 flights, re-optimizing them vertically and laterally, considering the following factors:

1.  Upper air wind and temperature
2.  Hi-resolution ash model data
25    3.  Detailed aircraft performance
4.  Cost of reduced engine lifetime due to ash damage
5.  Cost of increase in future fuel consumption due to ash damage and accumulation
6.  Effect of air density on ash mass ratio
7.  Effect of fuel flow on ash accumulation in engines (high fuel flow = high air flow = more ash accumulation)

30    It was demonstrated that the flight trajectories successfully avoided the ash cloud in the most economical way, even in complex ash-cloud / airway situations (airways only provide a very limited set of possible flight paths). Since maintenance cost rises sharply already with low ash concentrations, the ash cloud was often avoided entirely.

Analysis of the large-scale optimization run showed that 2% of the considered flights had to be considered as cancellation candidates due to high ash encounters, while 7% of the analysed flights required 4D-tailored trajectories. One important

conclusion was that the majority of flights were practically unaffected. The tailored trajectories that were applied to the flights were complex and tightly matched to a particular space/time configuration.

**6 Conclusion**

During the EUNADICS-AV exercise, it was demonstrated that tailored and selected observations as well as dedicated model
5    applications  successfully support aviation stakeholders in their decisions during an aviation crisis situation such as a dispersing volcanic ash cloud. A key objective of the exercise was to include a cost-benefit approach within the decision-making process.

One major result has been a presentation of the benefit of using a single harmonised portal for comprehensive visualisation of data and products. Such a platform enables crucial time-saving during a crisis, when decisions must be made fast and
10   efficiently. Once the data is visualised and processed, the possibility of using it integrated into flight modelling software enables a more effective decision-making process with the re-routing done in an automatic or quasi-automatic approach. For our particular case, despite the relevant event simulated and its proximity to the highly busy airspace, the exercise resulted in the surprising and useful results that most of the flights are unaffected and airspace is almost undisturbed. The estimated impact was therefore much lower than initially expected. For volcanic events, a cost-benefit approach was adopted, whereby
15   only flights above a certain threshold would be cancelled. The cost-benefit calculations revealed only for a small amount of cases that cancellation was an economically better option than the execution of the flight.

In the end, every air space closure or even re-routing of planes can immediately increase the costs for airlines which will lead to a certain risk acceptance, at least through regions which lie in an area with the ash concentration above the threshold. In future natural disasters, cost and disruption of air traffic can be eliminated to a great extent by combining dispersion models
20   with flight planning software to apply cost-based trajectory optimizations developed within the EUNADICS-AV project.

Such conclusions would not be possible if the working practices and interfaces between the broader natural hazard science communities on one side and the more application-oriented aviation community on the other side, mostly represented in the project by flight trajectory modelers and military aviation practitioners, would not have been established. The intense preparatory work ahead of the exercise managed to bridge existing gaps and bring together experts that have not cooperated
25   that closely before. This exercise, and the EUNADICS-AV project as a whole, has provided the very first steps towards integrating an impact-oriented perspective.

Furthermore, it was shown that even low aerosol concentrations have a significant impact on engine lifetime and future fuel consumption and therefore, justify continuous integration and optimization. Therefore, continuous monitoring, forecasting and optimizing for aerosols like ash, sand, dust, high altitude ice and sulphates, also during non-hazard situations, can
30   significantly reduce engine maintenance cost for airline operators and engine manufacturers.

[revised manuscript text omitted]

---

## Author Comment (AC1) · 4 Feb 2020

Dear Tanja! Thanks for your comprehensive revision of the paper and very good suggestions to improve the manuscript. I have uploaded a revised version which contains all your questions and comments (-> green, as well as the ones from the second reviewer Andreas Vogel (yellow)) and our answers and modifications in track change mode. Section 5 has been rewritten according to the reviewers recommendations. BR Marcus

Please also note the supplement to this comment:

[Figure]

https://www.nat-hazards-earth-syst-sci-discuss.net/nhess-2019-265/nhess-2019-265-AC1-supplement.pdf

**Supplement:**

**A volcanic hazard demonstration exercise to assess and mitigate the impacts of volcanic ash clouds on civil and military aviation**

Marcus Hirtl[1,16], Delia Arnold[1,2], Rocio Baro[1], Hugues Brenot[3], Mauro Coltelli[4], Kurt Eschbacher[5], Helmut Hard-Stremayer[6], Florian Lipok[7], Christian Maurer[1], Dieter Meinhard[7], Lucia Mona[8], Marie D. Mulder[1], Nikolaos Papagiannopoulos[8], Michael Pernsteiner[9], Matthieu Plu[10], Lennart Robertson[11], Carl-Herbert Rokitansky[5], Barbara Scherllin-Pirscher[1], Klaus Sievers[12], Mikhail Sofiev[13], Wim Som de Cerff[14], Martin Steinheimer[15], Martin Stuefer[16], Nicolas Theys[3], Andreas Uppstu[13], Saskia Wagenaar[14], Roland Winkler[15], Gerhard Wotawa[1], Fritz Zobl[5], Raimund Zopp[17]

[1]Zentralanstalt für Meteorologie und Geodynamik, Vienna, A-1190, Austria
[2]Arnold Scientific Consulting, Manresa, 08242, Spain
[3]Support to Aviation Control Service, BIRA-IASB, Brussels, B-1180, Belgium
[4]Osservatorio Etneo, Istituto Nazionale di Geofisica e Vulcanologia, Catania, 95125, Italy
[5]Department of Computer Sciences, University of Salzburg, Salzburg, 5020, Austria
[6]Kommando Streitkräfte/FachstabLu/J3(Lu), RefLtr Luftraumüberwachung, St. Johann im Pongau/Betriebsstelle Plankenau, 5600, Austria
[7]Brimatech Services GmbH, Vienna, A-1030, Austria
[8]Consiglio Nazionale delle Ricerche, Istituto di Metodologie per l'Analisi Ambientale (CNR-IMAA), Tito Scalo (PZ), 85050, Italy
[9]Joint Forces Command / Airstaff, Schwarzenbergkaserne, Wals, 5071, Austria
[10]CNRM, Université de Toulouse, Météo-France, CNRS, Toulouse, 31057, France
[11]Swedish Meteorological and Hydrological Institute, Norrkoping, SE-601 76, Sweden
[12]Klaus Sievers Aviation Weather, Lenggries, 83661, Germany
[13]Atmospheric Composition Research, FMI, Helsinki, FI-00101, Finland
[14]R&D Satellite Observations, KNMI, De Bilt, 3731 GK, Netherlands
[15]Austro Control GmbH, Vienna Int. Airport, Schwechat, 1300, Austria
[16]Geophysical Institute, University of Alaska Fairbanks, Fairbanks, AK 99775, USA
[17]Flightkeys, Vienna, A-1070, Austria

*Correspondence to*: Marcus Hirtl (marcus.hirtl@zamg.ac.at)

**Abstract.** Volcanic eruptions comprise an important airborne hazard for aviation. Although significant events are rare, e.g. compared to the threat of thunderstorms, they have a very high impact. The current state of tools and abilities to mitigate aviation hazards associated with an assumed volcanic cloud was tested within an international demonstration exercise. Experts in the field assembled at the Schwarzenberg barracks in Salzburg, Austria, in order to simulate the sequence of procedures for the volcanic case scenario of an artificial eruption of Etna volcano in Italy. The scope of the exercise ranged from the detection (based on artificial observations) of the assumed event to the issuance of early warnings. Volcanic emission concentration charts were generated applying modern ensemble techniques. The exercise products provided an important basis for decision making for aviation traffic management during a volcanic eruption crisis. By integrating the available wealth of data, observations and modelling results, directly into a widely used flight planning software, it was demonstrated that route optimization measures could be implemented effectively. With timely and rather precise warnings available, the new tools and

**Kommentiert [m1]:** VOGEL: Overstatement. Ice & water are more important due to higher frequency of occurrence and safety implications. Rephrase.

**Kommentiert [m2R1]:** DONE

**Kommentiert [m3]:** VOGEL: Slightly miss-leading to speak about detection as study is model based.

**Kommentiert [m4R3]:** DONE

[revised manuscript text omitted]

**Kommentiert [m6R5]:** Removed my sentence and added suggested text.

**Kommentiert [m7]:** VOGEL: General comment: The most critical impact is on the engine and it's components. Please describe this further as this is vital for a better understanding of the paper.

**Kommentiert [m8R7]:** See above

**Kommentiert [m9]:** VOGEL: Heroic? The restart of the engines can be explained by effects within the engine and the atmospheric and engine specific conditions. The air density and temperature differences between 11000 and 3600m as well as an increased core flow rate are main factors. The use of heroic is an overstretch.

**Kommentiert [m10R9]:** DONE

[revised manuscript text omitted]

hat formatiert: Absatz-Standardschriftart, Englisch (Vereinigtes Königreich)

hat formatiert: Englisch (Vereinigte Staaten)

Kommentiert [m19]: VOGEL: What is meant by this? Please clarify.

Kommentiert [m20R19]: DONE

Kommentiert [m21]: VOGEL: Please specify what you mean by "all the steps".

Kommentiert [m22R21]: DONE

Kommentiert [m23]: VOGEL: Please correct to LIDAR (in entire text) as it is an acronym

Kommentiert [m24R23]: DONE

Kommentiert [m25]: VOGEL: Related impacts of? Please specify.

Kommentiert [m26R25]: DONE

Kommentiert [m27]: VOGEL: Please specify further which one you mean.

Kommentiert [m28R27]: DONE

are repeated on a regular basis to test how procedures perform in real time situations. The exercises are  means to test communication networks and data exchange capabilities between the involved centres and groups. Important goals of such exercises are also checking the distribution of responsibilities and how seamless and coordinated necessary tasks are completed.

5  For volcanic ash, the annual European VOLCEX experiment  is the most important exercise with respect to volcanic ash and the impacts on aviation. The VOLCEX exercise involves the participation of VAACs, different airline operators, ANSPs, as well as different regulators and other crisis coordination cells. The objective of the exercise is to provide an opportunity for each individual state to test the effectiveness of their national crisis procedures, and for all the participants, their local volcanic ash contingency plans and procedures.

10  It is designed to test the operational capabilities and speed of all players involved in the industry (e.g. aircraft operators) that could be affected by a volcanic eruption in EU airspace. During the exercise, the crisis coordination between the various stakeholders via the European Aviation Crisis Coordination Cell (EACCC) and the Aircraft Operator Crisis Coordination Cell (AOCCC) is evaluated. ~~The VOLCEX exercise involves the participation of VAACs, different airline operators, ANSPs, as well as different regulators and other crisis coordination cells. The objective of the exercise is to provide an opportunity for 15 each individual state to test the effectiveness of their national crisis procedures, and for all the participants, their local volcanic ash contingency plans and procedures.~~

The EUNADICS-AV demonstration exercise was unique with respect to the abovementioned activities. The exercise did not only comprised all the items and timescales for a potential event relevant for aviation, but also looked at it from a research-oriented perspective. Innovative procedures, data and products were tested in a simulated environment. Such a complete and 20 comprehensive exercise demonstrated the applicability and feasibility of these innovative solutions into the aviation sector, produced by the research and operational capabilities of the EUNADICS-AV partnership. The exercise also demonstrated vast opportunities to support and complement the VAAC activities in the future e.g. by providing relevant observations, early warnings, source terms and analysis fields via a dedicated platform. This paper continues with describing the several activities during the exercise with focus on the volcanic test case and  the lessons learnt from 25 this multidisciplinary exercise.

**3 Overview of the EUNADICS-AV demonstration exercise set-up**

**3.1 General approach**

The exercise took place at the military facilities at the Schwarzenberg barracks from March 3rd to March 8th of 2019. The floor plan and location of the different exercise cells (Fig. 1) show the organisational aspects of the exercise and how the interaction 30 among cells was facilitated. Each cell was in charge of one of the pre-defined actions within a crisis situation (see Table 1 for details).
* * *
Margin annotations:

hat formatiert: Nicht Hervorheben

Kommentiert [m29]: VOGEL: (1) Structure a bit mixed up.
Kommentiert [m30R29]: DONE
Kommentiert [m31]: VOGEL: shift (1) to here
Kommentiert [m32R31]: DONE
Kommentiert [m33]: VOGEL: Which players do you mean?
Kommentiert [m34R33]: DONE

Kommentiert [m35]: VOGEL: (1) Structure a bit mixed up.
Kommentiert [m36R35]: DONE

hat formatiert: Nicht Hervorheben
hat formatiert: Nicht Hervorheben
hat formatiert: Nicht Hervorheben

Starting from (1) the detection (based on artificial observations) of the hazardous event and (2) the declaration of early warnings, (3) observations from different sources (e.g. satellite, LIDAR, in situ) were used to analyse the situation and were furthermore combined with models to determine the source terms and to refine the analysis (4). These results were then used to (5) cost-efficiently re-route airplanes. Every step of the whole procedure was executed with a demonstration of which data would be used in a real event and how the procedures and dependencies would take place (Fig. 2).

During the demonstration exercise, the cells presented the relevant data and impact on aviation during four phases of the event defined as:

- **Pre-alert**: aA notification of an event is received, which may lead to a possible major disruption of ATM (Air Traffic Management), requiring activation of an operational reaction chain.
- **Disruption**: Major disruption that impacts ATM operations and which may escalate to a crisis.
- **Crisis**: State of inability to provide air navigation service at required level resulting in a major loss of capacity, or a major imbalance between capacity and demand, or a major failure in the information flow following an unusual and unforeseen situation.
- **Recovery**: The operation will go back to normal and an evaluation of the impact will be finalized.

Each of the cells provided additional information on the used data sets and model applications as well as the scientific background of the procedures. Each cell was equipped with technical support, e.g., computers and monitors to demonstrate its role in the operational sequence of procedures and actions. Main results could be projected from each cell to various big screens in the hall (see Fig. 23).

Based on the observations and modelling data provided (see section 4) by the "scientific cells" (cell 2 to 6), the impact on aviation was simulated and depicted by the "aviation and stakeholder cells" (cell 7 to 15). These data allowed the ANSPs to release special aviation advisories (e.g. Significant Meteorological Information - SIGMETS). The impact of the event was recorded by keeping track of all aspects related to economically guided decision making in airline operational centres. Further impacts were due to re-routing and cancellation of flights by flight trajectory modellers (see section 5.3) as well as respective procedures invoked on the military side and participating airports and ministries. For the evaluation of the impacts on the air traffic, the air traffic was simulated by using the NAVSIM/ USBGSim (Rokitansky, 2009) simulator and was analysed by using Key Performance Indicators (KPIs). A newly developed cost model was used in the framework of an airline network balance tool (Flightkeys 5D) to cost-efficiently re-route flights affected by the disaster event (see section 5.3). Cell-specific EUNADICS-AV developments and impacts on ATM were shown for both, the civil and the military roles within each ATM phase.

The most important tasks in the preparation phase of the demonstration exercise, were to establish working practices and interfaces between the broader natural hazard science communities on one side and the more application-oriented aviation community on the other side. The latter was mostly represented by flight trajectory modellers and military aviation experts. experts. The emphasize at the EUNADICS-AV exercise was more on the scientific part and not on regular procedures, which

**Kommentiert [m37]:** BOLIC: Introduce the acroonym

**Kommentiert [m38R37]:** DONE

**Kommentiert [m39]:** VOGEL: Please define

**Kommentiert [m40R39]:** DONE

**hat formatiert:** Nicht Hervorheben

**Kommentiert [m41]:** BOLIC: This is a bit overstretching it as it was only simulated from one point of view.

**Kommentiert [m42R41]:** REMOVED

**Kommentiert [m43]:** BOLIC: You should mention how this is similar or different from the actual opertional setting in the ATM, and you should cite the ICAO volcanic ash procedures for the EU region https://www.icao.int/EURNAT/EUR%20and%20NAT%20Documents/EUR+NAT%20VACP.pdf

**Kommentiert [m44R43]:** DONE

**Kommentiert [m45]:** VOGEL: How is this different to regular procedure in case of a real volcanic eruption?

**Kommentiert [m46R45]:** DONE

are e.g. tested during the VOLCEX exercise where the crisis coordination between the various stakeholders through EACCC (the European Aviation Crisis Coordination Cell) and the Aircraft Operator Crisis Coordination Cell (AOCCC) is tested and evaluated. The intense preparatory work ahead of the demonstration exercise managed to bridge existing gaps. Experts such as natural hazard scientists to flight managers and pilots collaborated in an unprecedented fashion. This effectively was the spirit of the EUNADICS-AV project put into practice.

**3.1.1 The volcanic eruption scenario**

The exercise scenario was designed assuming an fictitious eruption of Etna volcano in Sicily, Italy. The period of the eruption was selected to guarantee an atmospheric circulation situation that favoured the transport of ash over large parts of Europe. The aim was to simulate the onset of the eruption during an episode with large scale weather patterns that led to a transport of ash from Sicily towards central Europe over a couple of days, with a wide spread ash cloud with concentration levels above the relevant thresholds (e.g. above 2 mg m$^{-3}$ at any vertical sub-column, see also section 1). A period lasting one week during April 2018 was chosen. The demonstration exercise used a fictitious Etna eruption starting on April 18, 2018, at 12:00 UTC with a plume height of 12 km above the vent height and a constant ash emission of 198 t/s. The eruption should last until April 20, 00:00 UTC. Following the initial volcanic source characteristics, tThe plume height and emission rate were further assumed to be with 10 km and 116 t/s until April 22, 00:00 UTC. The following days until April 25 were considered as the "recovery phase", when no relevant ash source was present over Europe anymore (see also section 4.3).

A fictitious event was chosen because there was no real eruption in the past that fulfilled the requirements which were defined for the exercise, e.g. that the volcanic ash cloud spread over Europe with certain threshold exceedances of volcanic ash concentrations over selected regions (e.g. over Austria where ANSP have to deliver specific products).

**3.1.2 Data sharing and visualization**

All data sets (artificial observations and model data, see section 4) used for the demonstration exercise were accessible and visualized on a dedicated portal. The information flow is depicted in Fig. 34. The various data products (satellite and ground based observations, modelling data) that were requested by the wide range of users, had many different data sources, data types and formats, projections, sampling time intervals, and coverage.

The EUNADICS-AV project made use of existing data channels and protocols to provide a harmonised and easily accessible portal (see Fig. 54 for example) for all the different types of information, including observations or modelling results. The portal allowed the participants of the exercise to explore the event and the products available anytime during each of the four phases with a graphical user interfacevisually attractive and easy to use interface.

**Kommentiert [m47]:** VOGEL: a fictitious

**Kommentiert [m48R47]:** DONE

**Kommentiert [m49]:** VOGEL: Please explain where this value is coming from and why is was chosen. Why is it such an important value for aviation? Bear in mind that this is an European view

**Kommentiert [m50R49]:** DONE

**Kommentiert [m51]:** BOLIC: why this threshold? Based on what? You need to specify why took this value. Also take into account that only EUropean VAACs are using it.

**Kommentiert [m52R51]:** DONE

**Kommentiert [m53]:** VOGEL: Subjective statement. Either delete or mention "with a graphical user interface"

**Kommentiert [m54R53]:** Added to 3.1.1

**4 Datasets used for the demonstration exercise**

**4.1 Artificial observations**

As the considered Etna scenario was artificial, the observational data sets were generated based on model simulations. The task of creating the artificial observations splits to two steps: (i) simulate the evolution of the artificial volcanic plume, (ii) simulate the fingerprint of this plume for different types of observation devices.

**4.1.1 Simulations of the artificial plume evolution**

Having the parameters of the artificial eruption decided, we have run the SILAM (System for Integrated modeLling of Atmospheric composition) modelling system of FMI (http://silam.fmi.fi, Sofiev et al, 2015) was run over the European domain simulating the dispersion of the emitted masses in the atmosphere (see example in Fig. 65, left).

The main challenge at this stage was to simulate not only the main dispersion but also account for incompleteness of our knowledge of the atmosphere, e.g. its dynamics during these days, actual wind direction and speed, etc.. To account for this uncertainty, the SILAM model was not run with a realistic meteorologicaly data from during the selected period but the fields were taken not from the operational weather models but rather from the reanalysis dataset ERA-5 (Hersbach and Dee, 2016) of the European Centre of Medium-Range Weather Forecast (ECMWF). ERA5 was produced by assimilating vast amounts of historical observations into a numerical model and provides therefore high-quality meteorological data on a global scale, the final data set will extend back to 1950. Compared to the operational weather predictions, the re-analysis has noticeably more extensive assimilation capabilities (see summary for the previous version of the ECMWF reanalysis ERA-Interim in (Dee et al., 2011) and thus can be considered as a more accurate representation of the actual meteorological conditions than the operational forecasts. All other simulations of with SILAM and other models used only operational forecasts for the corresponding period.

**4.1.2 Generation of artificial observations from SILAM simulations**

The output of the plume dispersion simulations consisted of the 4-D distribution of the ash and $SO_2$ concentrations. To obtain the artificial observations, we applied the corresponding observation operators, which "observed" the concentration distribution as if the corresponding device would have done it. For instance, for the in-situ sampling observations, the concentrations near-surface were extracted at the locations of the stations. For EARLINET (European Aerosol Research Lidar Network; Pappalardo et al., 2014; www.earlinet.org) EARLINET lidarsLIDAR, the vertical profiles of concentrations over the lidar LIDAR locations were convoluted with their sensitivity. For satellites, the averaging kernels of the instruments were convoluted with the vertical profiles of concentrations along the satellite trajectory, etc. We also took into account the actual weather impact and instrument specifics: as shown in the example in Fig. 56 (right), IASI (Infrared Atmospheric Sounding Interferometer, e.g. Karagulian, 2010) instrument cannot observe in cloudy conditions and cannot retrieve very thick ash layers. At the final stage, the artificial plume retrievals were summed-up with the actual satellite data for the corresponding orbits.

**Kommentiert [m55]:** VOGEL: What was the decision behind using simulated plume heights rather than sing data from historic Etna eruption? Please add.

**Kommentiert [m56R55]:** DONE

**Kommentiert [m57]:** BOLIC: A word on why simulation and not some historical event would be welcome. I.e. could not find such an event in the past...

**Kommentiert [m58R57]:** DONE

**Kommentiert [m59]:** VOGEL: Acronym for? Not defined yet.

**Kommentiert [m60R59]:** DONE

**Kommentiert [m61]:** VOGEL: What do you mean by realistic? Please explain.

**Kommentiert [m62R61]:** DONE

**Kommentiert [m63]:** VOGEL: Explain why and what it is. Or add reference.

**Kommentiert [m64R63]:** DONE

**Kommentiert [m65]:** VOGEL: Mentioned earlier. Please define when first used.

**Kommentiert [m66R65]:** First time here

**Kommentiert [m67]:** VOGEL: Define or add reference.

**Kommentiert [m68R67]:** ADDED

**Kommentiert [m69]:** VOGEL: Explain or add reference.

**Kommentiert [m70R69]:** DONE

**4.2 The Early Warning System (EWS)**

The purpose of the EWS is to provide near real-time (NRT) observational data in case of the detection of an airborne hazard. Subsequently, it provides centralized information for NRT monitoring, to contribute to the improvement of the analysis and forecasts of volcanic ash plumes.

5 Observational data (satellite, ground based and airborne remote sensing and in-situ) play a significant role to determine the 4D distribution of the ash cloud. For the demonstration exercise, several ground- and space-based observations (synthetic) were chosen to facilitate the detection of the event (see in sections 4.2.1 to 4.2.3) and to assess the current extent and location of the dispersed ash cloud.

During the event, alerts created from these synthetic observations were delivered to the different cells to trigger their respective 10 actions. As in a quasi-operative mode, with NRT products derived from the available monitoring networks, appropriate information from notifications of the hazardous event were provided.

**4.2.1 Volcano observatory Sicily**

The Etna Volcano Observatory of Italian National Institute of Geophysics and Volcanology (INGV) produced five VONA (Volcanic Observatory Notice for Aviation) messages for the artificial Etna case. The first VONA message was YELLOW, 15 indicating that the volcano showed signs of elevated unrest above its background level (6:00 UTC on April 18, 2018). Then, 10 minutes after the start of Etna eruption (at 12:00 on April 18, 2018), a RED message informed that lava fountaining started from the central crater summit vent, with large ash emission occurring up to 122000 m above the sea level (a.s.l.) and the ash plume moving towards the north. At 00:10 UTC on April 20, 2018, a second RED message was provided. This message confirmed the on-going eruption, with ash emissions up to 10000 m (a.s.l.). Just after 00:00 UTC on April 23, 2018, an 20 ORANGE message indicated the end of lava fountaining and ash emission. Next, a GREEN message followed, with the announcement that the volcanic activity has ceased (volcano reverts to its normal/non-eruptive state) and that no ash cloud was produced anymore.

**4.2.2 Synthetic EARLINET/ACTRIS data**

The European Aerosol Research Lidar Network (EARLINET; Pappalardo et al., 2014; www.earlinet.org), established in 2000, 25 provides aerosol profiling data on a continental scale. EARLINET is part of the Aerosols, Clouds, and Trace gases Research InfraStructure (ACTRIS; www.actris.eu). The main EARLINET products are the aerosol backscatter and extinction profiles. The data allowed investigating several volcanic eruptions in the last 20 years (e.g. Etna 2002 eruption; e.g. Pappalardo et al., 2004; and Eyjafjallajökull volcano in Iceland Pappalardo et al., 2013).

Within the EUNADICS-AV project, an EWS was designed that relies explicitly on EARLINET aerosol observations. This 30 product is based on the possibilities offered by the EARLINET Single Calculus Chain (SCC; D'Amico et al., 2015) for the NRT data processing and the generation of tailored products network-wide. The calibrated high resolution lidar data serve as
* * *
**Kommentiert [m71]:** VOGEL: see chapter 3.11

**Kommentiert [m72R71]:** First days 12 km then 10 km

**Kommentiert [m73]:** VOGEL: General comments for this sub-chapter including plots:

Perhaps combine Fig. 6 & 7 for on station only e.g. Leipzig. For example show three sub-figures.
1. Backscatter coefficient when cloud cover is removed.
2. Backscatter coefficient when cloud cover is not removed.
3. Alert for aviation plot

That would make the understanding a bit easier.

**Kommentiert [m74]:** VOGEL: Why is Fig. 7 left not correlated with right, especially on 23/04 above 15 km. Please explain.

**Kommentiert [m75]:** VOGEL: Mentioned earlier. Please define when first used.

**Kommentiert [m76]:** VOGEL: Generally too much information in this paragraph compare to others. A single sentence plus a reference would be good for a better readability.

**Kommentiert [m77R76]:** Removed the last part

the basis for the alert delivery (Baars et al., 2017). This EWS provides only qualitative and not quantitative information thus the EARLINET EWS represents a warning system rather than a tool for decision makers. The demonstration exercise was the first occasion  for which the EARLINET EWS was tested.

The EARLINET products for the demonstration exercise consisted of snap-shots of the  LIDAR signals and of the EWS plot. Figure 7 shows an example of the particle backscatter coefficient (with the cloud fraction removed and when not) for the EARLINET station in Barcelona. Special attention should be given to the particle backscatter coefficient values reported in the figures as the values are unrealistically high and most likely would attenuate the  LIDAR laser beam.

The simulated data refers only to volcanic ash and the depolarization information was not incorporated in the alert delivery method.  Figure 8 presents an example of the attenuated backscatter coefficient and the corresponding alert for aviation for the EARLINET Leipzig station. The ash layer first appears at around 13 km on April 22, 05:00 UTC, with high values reaching the ground the next day. The ash cloud is not visible anymore over the Leipzig EARLINET site on April 24.

**4.2.3 Synthetic satellite data simulated for IASI and MODIS**

Satellite observations from 2 types of sensors (IASI and the Moderate-resolution Imaging Spectroradiometer (MODIS)) were considered for generating alert products for the exercise. Clerbaux et al. (2009) and Levy et al. (2015) gave descriptions of the IASI and MODIS instruments respectively. Synthetic ash products from two IASI sensors (onboard MetOp-A and MetOp-B; Clarisse et al., 2010) and aerosol optical depth (AOD) from two MODIS sensors (onboard Aqua and Terra) were used.

After the detection of the aerosol/ash cloud at the Nicolosi and Catania EARLINET stations, the first selective detection of ash from satellite was created for IASI-A (Fig. 8, left) few ash pixels at 20:10 UTC on April 18, 2018). A first partial detection was created for 08:28 UTC on April 19, 2018, followed by a global detection of the ash plume by IASI-B (Fig. 8, right) at 09:27 UTC on April 19, 2018. Following each of these synthetic detections a warning was issued with a time delay of about 1.5 hours.

Synthetic observations of the ash plume (AOD anomalies) were created for the MODIS instruments aboard of the Terra and Aqua satellites in similar fashion as for IASI-A and B. The MODIS detections were selected with respective daily timestamps of 10:19 and 11:58 UTC. Warnings were delivered to the aviation and military cells in support of their decision, actions and tasks. The warnings were also used to provide simulation start times for the dispersion models, improving the capability for achieving advanced analysis and forecasts of Etna's ash cloud.

**4.3 Model ensemble**

The following models provided the concentration distribution for the whole period using the pre-defined source term: MATCH (Multi-scale Atmospheric Transport and Chemistry, Robertson et al., 1999), MOCAGE (Modèle de Chimie Atmosphérique de Grande Echelle, Guth et al., 2016), SILAM (Sofiev et al., 2015), FLEXPART (FLEXible PARTicle dispersion model, Stohl et al., 2005) and WRF-Chem (Weather Research and Forecasting (WRF) model coupled with Chemistry, Grell et al., 2005; Stuefer et al., 2013). Although the EUNADICS-AV partnership used a reduced number of models, this approach accounts for

**Kommentiert [m78]:** VOGEL: Please expand on the statement of with/without cloud fraction. You show the difference, but you are not describing the difference etc.

**Kommentiert [m79R78]:** The EWS is based on signals with no clouds, so the cloud removal before applying the method is fundamental. In order to keep short the paragraph, as requested from the reviewer, we included just one case as example after cloud screening and EWS. There is no need to show the one with clouds in this paper: the relevance of the cloud screening will be described in the paper about this product.  Additionally the apparent missing of correlation between the backscatter temporal evolution and EWS is due to the choice color scale of the plots: they are set on the base of realistic values, while here the observations /(as stated before) are really unrealistic. However, in order to avoid misunderstanding a different scale is reported in the revised version of the paper. The structure visible above 15 km is related to aerosol presence as simulated in the exercise.

**Kommentiert [m80]:** VOGEL: Please explain or add reference.

**Kommentiert [m81R80]:** DONE

an ensemble of multiple models. The 4D-model values were interpolated to a common grid, which allowed to calculate a mini-ensemble and percentiles indicating the model uncertainty for the considered time step and location. For the demonstration exercise, the 75th percentile ash concentration level was used, which corresponds to ash concentrations below 75 % of the modelled outputs. Using this approach was a slightly conservative compromise as the median (50th percentile) is the most probable scenario. Using the 75th percentile means that the regions that lie above a certain threshold are larger than for the median..

Figure 109 shows the dispersion of the volcanic ash cloud over several days at a selected layer (FL275). Note that the model data were produced in advance of the demonstration exercise. The modelled results were used as baseline for the aviation related tasks. The data wasmodelled data were interpolated to 13 flight levels, visualized on the EUNADICS-AV portal and also imported into other applications like flight planning software and Vvisual Wweather (https://www.iblsoft.com/products/visualweather/).

**5 The impact of the ash cloud on aviation for the Etna eruption scenario**

**5.1 Air Navigation Service Provider (ANSP)**

At this demonstration exercise, the tasks designated to ANSP's during such a crisis situation were executed by Austro Control (ACG). Aviation weather servicesACG provided specially tailored warnings and/or products for different aviation stakeholders in order to warn about the presence of the volcanic ash cloud. During the course of the demonstration exercise, the datasets described in the previous sections were fed into the visualisation software that is also used for daily operations. Subsequently, products were generated for the local situation in Austria in order to give the exercise participants an impression of available aviation products for the case of a volcanic ash eruption. The presented products included internationally prescribed and harmonized weather warnings or prediction products, as well as country specific products for Austria. In addition, a pilot briefing was conducted including specific information on the volcanic ash event for a potentially affected flight.

The following internationally harmonized products (selected examples) were prepared (according to an ACG internal guideline):

- Volcanic ash SIGMET (Significant Meteorological Information)

  A SIGMET information is a warning message of the occurrence or expected occurrence of specified en-route weather phenomena, which may affect the safety of aircraft operations (ICAO, 2018a). MWO Vienna issued a volcanic ash SIGMET for April 22, 2018, at 16:00 UTC. It describes that a volcanic ash cloud originated from Mount Etna; the cloud is predicted at 17:00 UTC between FL250 and FL350 in the Austrian airspace within the polygon of which the longitude and latitude coordinates are given (grey line in Fig. 110). Around 18:00 UTC, the ash cloud spread to the southeast and a second polygon is provided (in red in Fig. 110). These polygons were generated from the 75[th] percentile of the ensemble model output (see section 4.3).

- Volcanic ash NOTAM (Notice to Airmen)

**Kommentiert [m82]:** VOGEL: Vague statement. Please explain what you mean or what your definition of conservative compromise is.

**Kommentiert [m83R82]:** DONE

**Kommentiert [m84]:** VOGEL: Explanation needed. No word of EARLINET mentioned, but the figure caption of Fig. 9 says EARLINET

**Kommentiert [m85R84]:** Adjusted figure caption, removed EARLINET

**Kommentiert [m86]:** VOGEL: Already defined.

**Kommentiert [m87R86]:** ADJUSTED

[revised manuscript text omitted]

30    were used for further calculation of suitable deviation routes. The re-calculated routes were simulated and visualized by
* * *
[1] European Civil Aviation Conference (ECAC) currently includes 44 member states (refer to https://www.ecac-ceac.org/member-states).

Kommentiert [m94]: BOLIC: I would say that the cloud was depicted or something of the sort. Figure 12 shows the cloud position over the sectors boundaries and airport positoins, aircraft position and nothing else.

Kommentiert [m95]: VOGEL: Please define.

Kommentiert [FZ96R95]: DEFINED par. below, no abbrev.

Kommentiert [m97]: BOLIC: As these are not mentioned anywhere else in the text, you should delete the references to the KPIs

Kommentiert [FZ98R97]: DELETED

Kommentiert [m99]: VOGEL: In what time period? For Airliners and Business aircraft as well? Military flights?

Kommentiert [FZ100R99]: DONE

Kommentiert [m101]: BOLIC: A bit more details on the number of flights - are they taken for the days that were simulated int he exercise, which data from eurocotnrol?

Kommentiert [FZ102R101]: DONE

NAVSIM (3 seconds time step). Selected flights were simulated in real time using the Laminar Research's flight simulator (X-Plane 11, 2019) and were visualized on-line in NAVSIM/ USBGSim. Furthermore, voice communications concerning visually observed volcanic ash were exchanged in real time between involved "humans in the loop" respectively a pilot and an air traffic controller.

5    A newly developed cost model was used in the framework of an airline network balance tool (Flightkeys 5D, https://www.airlinesoftware.net/product/1422/flightkeys-5d) to cost-efficiently re-route flights affected by the disaster event. A realistic damage cost model has been developed in close cooperation with with the advice of the aircraft engine manufacturer Rolls-Royce aircraft engine manufacturer, and the 5D Flightkeys trajectory optimizer was modified to integrate the cost model and 5D² 5D ash data. ATM flow restrictions and mandatory routes were disabled to reduce complexity and to allow more

10   efficient re-routings. Since the vast majority of flights were unaffected and could be assumed to proceed according to their originally filed routings, network disturbance was assumed to manageable at a level that would be similar to a large convective weather situation. Furthermore, the assumption was made that a future Europe-wide ATM system would be capable to accommodate the rerouting requests in an efficient way for the case where practically all airspace users would utilize the advanced trajectory planning capability. It can be validly assumed that at the initial stages of the introduction of such

15   optimization tools the "early adopters" would gain additional benefit from the fact that most other airspace users would have to cancel their affected flights and the most affected airspaces would thus be generally less utilized.
The "domino" effect of delayed or cancelled flights on connecting flights was not simulated in the exercise. A conservative guess on that effect would be a doubling of the predicted flights cancellation rate.
The Flightkeys system during the EUNADICS-AV exercise is shown in Fig. 13. Hi-resolution volcanic ash data for the entire

20   scenario period (April 18 to 25, 2019) was imported as 75 percentile polygons (83000 polygons) with a temporal resolution of 1 hour. Polygons were imported for the following contour levels intervals: <1, 1=<2, 2=<3, 3=<4. >=4 1.0, 2.0, 3.0, >=4.0 mg/m3. A large-scale, flight by flight, optimization was performed on the entire set of 98000 flights, re-optimizing them vertically and laterally, considering the following factors:

1. Upper air wind and temperature (GRIB data in 6h time resolution, 1.25° lateral and 2000ft vertical)
25  2. Hi-resolution ash model data (see above)
3. Detailed aircraft performance (not BADA model, but OEM-provided flight planning data)
4. Cost of reduced engine lifetime due to ash damage
5. Cost of increase in future fuel consumption due to ash damage and accumulation
6. Effect of air density on ash mass ratio
* * *
² In addition to the 4D space covered by traditional flight planning solutions, 5D extends the calculation space into a 5th dimension. Uncertainties in surface weather, traffic and cost prediction is modelled into statistical functions based on a continuous analysis of actual flight data. For upper air weather, multicasting weather products are introduced to compare multiple scenarios and automatically apply suitable strategies, e.g. adaptive fuel reserves and delay cost reduction (for more information see www.flightkeys.com/).
* * *
**Kommentiert [m103]:** VOGEL: The damage cost model was not developed in close cooperation with Rolls-Royce. Rolls-Royce did advise Flightkeys, but was not involved in any model development. Please re-write.

**Kommentiert [m104R103]:** Re-written

**Kommentiert [m105]:** BOLIC: What is the 5th dimension? It is not explained anywhere.

**Kommentiert [m106R105]:** See footnote

**Kommentiert [m107]:** BOLIC: Other important things not taken into account are the individual airline Safety Risk Assessment - that is to say what decisions they take when ash is forecasted - a significant percentage of airlines would not fly in any forecasted concentration of ash. Further, the connections between flights are not taken into account - if an aircraft that is supposed to perform the flight does not arrive (or does not arrive in time), then the next flight will not happen (or will be delayed). It also does not say anything if the simulator keeps the aircraft separated (I assume it does, but it is not mentioned). Several of these points have already been mentioned several times, and they make the current conclusions invalid!!!

**Kommentiert [m108R107]:** See new text

**Kommentiert [m109]:** VOGEL: Do you mean
< 1
1-2
2-3
3-4
> 4 mg/m3
?

**Kommentiert [m110R109]:** Changed

**Kommentiert [m111]:** BOLIC: Was it a network-wide optimisation, or flight by flight trajectory optimisation? My understanding is that the trajectory planner optimises on flight by flight basis - looking for the cost-optimum for a trajectory, but I might be wrong. The kind of optimisation has to be specified.

**Kommentiert [m112R111]:** Flight by flight

**hat formatiert:** Hochgestellt

7. Effect of fuel flow on ash accumulation in engines (high fuel flow = high air flow = more ash accumulation)

It was demonstrated that the flight trajectories successfully avoided the ash cloud (see examples in Fig. 14 & Fig. 15) in the most economical way by applying a newly developed algorithm that predicts future maintenance cost and fuel efficiency losses, even in complex ash-cloud / airway situations (airways only provide a very limited set of possible flight paths).

The algorithm is based on the assumptions that air flow through the engine core is proportional to fuel flow and thus allows a direct correlation of ash accumulation to fuel flow. Rolls-Royce contributed first estimated ratios of ash mass accumulation versus engine deterioration (e.g. exhaustgas-temperature margin loss per kg of ash), thus allowing to predict both engine efficiency and engine lifetime decreases and their cost equivalents. The direct relationship to fuel flow is extremely well suited to the already well-established cost optimization algorithm of FLIGHTKEYS' flight planning system.

With full availability of free-route airspaces across the entire ECAC airspace, even better avoidance trajectories can be expected. Since maintenance cost rises sharply already  at low ash concentrations, the ash cloud was often avoided entirely. Analysis of the large-scale optimization run showed that 2% of the considered flights had to be considered as "hard" cancellation candidates as a direct consequence  of excessive  ash  concentrations along their routes, while 7% of the analysed flights required 4D-tailored trajectories. As mentioned above, a conservative propagation estimate of cancellations due to aircraft or crew rotation aspects would probably lead to a doubling of that cancellation rate to 4% which is still very low. One important conclusion was that the majority of flights were practically unaffected.

**6 Conclusion**

During the EUNADICS-AV exercise, it was demonstrated that tailored and selected observations as well as dedicated model applications can successfully support aviation stakeholders in their decisions during an aviation crisis situation such as a dispersing volcanic ash cloud. A key objective of the exercise was to include a cost-benefit approach within the decision-making process.

One major result has been a presentation of the benefit of using a single harmonised portal for comprehensive visualisation of data and products. Such a platform enables crucial  timesaving during a crisis, when decisions must be made fast and efficiently. Once the data is visualised and processed, the possibility of using it integrated into flight modelling software enables a more effective decision-making process with the re-routing done in an automatic or quasi-automatic approach. For our particular case, despite the relevant event simulated and its proximity to the highly busy airspace, the exercise resulted in the surprising and useful result that by integrating ash contamination effects into cost-based trajectory optimization algorithms, most of the flights are almost unaffected and remaining airspace is much better utilized than  . The estimated impact was therefore much lower than initially expected. For volcanic events, a cost-benefit approach was adopted, whereby only flights above a certain threshold would be cancelled. The cost-benefit calculations revealed only for a small  number of cases that cancellation was an economically better option than
* * *
**Kommentiert [m113]:** VOGEL: Please expand. It feels like that this is one of the novel aspects of this exercise and paper. Therefore, information behind each of the listed factors would help to better understand the optimisation. What assumptions went into these 7 factors.

**Kommentiert [m114R113]:** See in the list and below in the text

**Kommentiert [m115]:** VOGEL: Is there a good way to illustrate the trajectories? That would allow the reader to better understand the specific scenario.

**Kommentiert [m116R115]:** See Fig. 14&15

**Kommentiert [m117]:** BOLIC: I would describe here that the inclusion of the maintenance cost in the trajectory optimisation has an important impact on automatically moving the flight away from the cloud. As that is I think the most important point here.

**Kommentiert [m118R117]:** See modified text

**Kommentiert [m119]:** BOLIC: And are usually not used in the critical situations, as the best tactical re-routing is sought.

**Kommentiert [m120]:** VOGEL: What was the criteria for the calculation? All the above factors? Please explain.

**Kommentiert [m121R120]:** See modified text

**Kommentiert [m122]:** VOGEL: Context of this sentence is missing. Please take out or add more information.

**Kommentiert [m123R122]:** Removed the sentence

**Kommentiert [m124]:** BOLIC: 2% of flight cancellations is highly optimistic - this is just counting the particular flights that would not be able to fly, but in the real life these bring along other flights that were supposed to be performed with the same aircraft. If the aircraft does not show up, then the next flight is also cancelled even if it could be performmed. Based on all the things not taken into account by the simulation (most important ATM capacity and flow, and the airlines' Safety Risk Assessment) you CANNOT conclude that the majority of flights were practically unaffected. You could say that in some future scenario where all the airlines would fly through some concentration of ash, the ATM capacity is available in unconstrained quantities, then those flights would be unaffected. ...

**Kommentiert [m125R124]:** See modified text

**Kommentiert [m126]:** VOGEL: Which is unfortunately not shown.

**hat formatiert:** Nicht Hervorheben

**Kommentiert [m127]:** VOGEL: Slightly misleading statement. A key conclusion of this exercise is the fact that ...

**Kommentiert [m128]:** BOLIC: This is a false conclusion based on all the previous comments. What is really importan ...

the execution of the flight. The cancellation threshold was set at additional cost exceeding 200% of all other considered operating costs (fuel, time, overflight charges).

In the end, every air space closure or even re-routing of planes can immediately increase the costs for airlines which will lead to a certain risk acceptance, at least through regions which lie in an area with the ash concentration above the threshold. During the exercise we have shown for a volcanic ash scenario, that  cost and disruption of air traffic can be eliminated to a great extent by combining dispersion models with flight planning software to apply cost-based trajectory optimizations developed within the EUNADICS-AV project.

It can be disputed how big such positive effects would be in the current, fragmented and over-regulated European airspace, but this only underlines how important future progress in the automation and unification of ATM systems and processes will be to allow more flexibility in airspace disruption scenarios like the ones simulated in the exercise

Such conclusions would not be possible if the working practices and interfaces between the broader natural hazard science communities on one side and the more application-oriented aviation community on the other side, mostly represented in the project by flight trajectory modelers and military aviation practitioners, would not have been established. The intense preparatory work ahead of the exercise managed to bridge existing gaps and bring together experts that have not cooperated that closely before. This exercise, and the EUNADICS-AV project as a whole, has provided the very first steps towards integrating an impact-oriented perspective.

**7 Acknowledgements**

This work has been conducted within the framework of the EUNADICS-AV project, which has received funding from the European Union's Horizon 2020 research programme for Societal challenges - smart, green and integrated transport under grant agreement no. 723986. We thank the Austrian military for hosting the exercise and providing technical equipment together with the University of Salzburg.

**Kommentiert [m129]:** VOGEL: Based on what assumption?

**Kommentiert [m130]:** VOGEL: This exercise was based on volcanic ash injection and distribution into the atmosphere. Other contaminants, e.g. sulphur dioxide, can effect decisions, engine health etc. quite differentially. To say that cost and disruptions of air traffic can be eliminated is wrong and not serious.

**Kommentiert [m131R130]:** Rewritten

**Kommentiert [m132]:** BOLIC: Cannot be eliminated! The sofware can be used in such events, true. Would probably bring benefits, but only if all the usual constraints are first introduced in the software. If we had unlimited capacity in the ATM we would not have any delays when we get bad weather for example, and that is esentially what this particular simulation was based on - no constraints.

**Kommentiert [m133R132]:** See comment below

**Kommentiert [m134]:** VOGEL: This was not mentioned or shown in the paper at all. Either take out this statement or add low concentration examples as well.

**Kommentiert [m135R134]:** REMOVED

**Kommentiert [m136]:** BOLIC: This is a very important conclusion, and should be expanded rather than the previous one. There is a need for quantification of the engine lifetime and maintence costs that currently does not exist (at least not publicly)

**Kommentiert [m137R136]:** According to Rev2 I have removed it

**Kommentiert [m138]:** VOGEL: True statement, nut has nothing to do with the presented study. Better take it out.

**Kommentiert [m139R138]:** Removed

[revised manuscript text omitted]

**Kommentiert [m142]:** VOGEL: Acronym used in the text, but not explained.

**Kommentiert [m143R142]:** DONE

**Kommentiert [m144]:** VOGEL: See above statement.

**Kommentiert [m145R144]:** DONE

**Kommentiert [m146]:** VOGEL: See above statement.

**Kommentiert [m147R146]:** DONE

NRT          - Near Real Time

OEM          - Original Equipment Manufacturer

PIREP          - Pilot Report

PMA          - Parts Manufacturer Approval

SCC          - Single Calculus Chain

SIGMETS          - Significant Meteorological Information

SILAM          - System for Integrated modeLling of Atmospheric coMposition

STC          - Supplemental Type Certificate

VAACs          - Volcanic Ash Advisory Centres

VACP          - Volcanic Ash Contingency Plan

VOLCEX          - VOlcanic Ash Contingency EXercise

VONA          - Volcanic Observatory Notice for Aviation

WMO          - World Meteorological Organization

WRF-Chem          - Weather Research and Forecasting (WRF) model coupled with Chemistry

**Kommentiert [m148]:** VOGEL: See above statement.

**Kommentiert [m149R148]:** DONE

**Figures**

[Figure]

Figure 1: Top: floor plan of different cells at Schwarzenberg barracks (Salzburg, Austria). Bottom: Picture of the main hall of the demonstration exercise premises.

[Figure]

[Figure]

**Figure 2: Selected photos from the demonstration exercise at the Schwarzenberg barracks in Salzburg. The flight simulator was operated by both military and civil pilots.**

[Figure]

5    **Figure 3: EUNADICS-AV exercise 2019 workflow.**

hat formatiert: Englisch (Vereinigtes Königreich)

hat formatiert: Englisch (Vereinigte Staaten)

[Figure]

**Figure 43: Information flow between data providers and users.**

[Figure]

**hat formatiert:** Schriftart: (Standard) +Textkörper (Times New Roman), 11 Pt., Schriftfarbe: Benutzerdefinierte Farbe(RGB(0;0;10))

[Figure]

**Figure 45: Exemplified visualization of the EUNADICS-AV portal showing the model ensemble (a) and the vertical ash distribution (b) at an EARLINET/ACTRIS station (Barcelona) on April 21, 2018.**

**Kommentiert [m150]:** VOGEL: Please add numbering system of sub-plots (e.g. a), b),...

[Figure]

**Figure 56: Left: Simulated observations based on actual IASI pixels; no retrievals if: cloud cover fraction > 0.25 in the 0.1° x 0.1° model pixel. Right: simulated AOD > 8. The values are perturbed with a spatially correlated error (the spatial correlation is based on a similar error covariance as the simulated MODIS AOD)-the error of each data point is set to 0.25 times the simulated value plus 0.05 g m-23.**

[Figure]

**Figure 67: Particle backscatter coefficient time-height evolution when the cloud fraction is removed (left panel) and when not (right panel) for the Barcelona EARLINET station and for the whole exercise period.**

[Figure]

**hat formatiert:** Englisch (Vereinigte Staaten)

**Figure 78: The attenuated backscatter coefficient (left panel) and the alert for aviation (right panel) for the Leipzig EARLINET station.**

[Figure]

**Figure 98: Warnings of ash plume by IASI-A (at 08:28 UTC) and IASI-B (at 09:27 UTC) on April 19, 2018.**

[Figure]

**Figure 109: Model ensemble (75-percentile)  for the 6 consecutive days after the artificial Etna eruption in April 2018. The data is depicted on FL275.**

[Figure]

Figure 11: Location of the two polygons for which the SIGMETS (ACG) were issued.

[Figure]

Figure 12: Low-level significant weather chart for April 23, 14:00 UTC. The predicted volcanic ash is represented by the brown dash-dot-dotted line, over the Adriatic Sea, parts of Italy and Slovenia, the south-eastern Switzerland and the southern half of Austria and Hungary extends from the surface to over FL250.

[Figure]

[Figure]

**Figure 13:: Flightkeys system during the EUNADICS-AV exercise. Left: overview of all monitored flights. Right: sStatus (location, height, planned route) of an individual flight.**

Kommentiert [m151]: VOGEL: Please offer more explanation to better understand the plot.

Kommentiert [FZ152R151]: ADDED

[Figure]

[Figure]

**hat formatiert:** Schriftart: Kursiv

**Figure 12̶34: Volcanic ash** *(red: >=4 mg/m3, yellow, 2-4 mg/m3, green 0.2-2 mg/m3)* **reroute calculated by FlightKeys and simulated with NAVSIM̶/ USBGSim (University of Salzburg)**, laterally and vertically avoiding ash concentration.

**Kommentiert [m153]:** VOGEL: Please offer more explanation to better understand the plot.

**Kommentiert [FZ154R153]:** ADDED

[Figure]

Figure 15: Example of a rerouting of one flight calculated by FlightKeys and simulated with NAVSIM/ USBGSim (University of Salzburg), avoiding laterally and vertically vulcanic ash (red: >=4 mg/m3, yellow, 2-4 mg/m3, green 0.2-2 mg/m3 in different FLs).

hat formatiert: Schriftart: (Standard) +Textkörper (Times New Roman), Kursiv, Schriftfarbe: Benutzerdefinierte Farbe(RGB(0;0;10))

Kommentiert [m155]: VOGEL: Please offer more explanation to better understand the plot.

Kommentiert [FZ156R155]: ADDED

Kommentiert [FZ157]: MH please check values

**Tables**

Table 1: EUNADICS-AV demonstration exercise cells and related tasks.

**Kommentiert [m158]:** VOGEL: Table 1 not mentioned in text. Please explain content of the table in text.

| No. | Cell name | Tasks |
|---|---|---|
| C 1 | **EUNADICS-AV Exercise Lead** | Management team of EUNADICS-AV demonstration exercise (was located in an extra room, not depicted here). |
| C 2 | **Observational Infrastructure** | Collection, tailoring, documentation, and distribution of observational data such as satellite and lidar data. |
| C 3 | **Early Warning System** | Sending of early warning messages and triggering of the EUNADICS-AV modelling chain based on the observational data provided by C 2. |
| C 4 | **Data integration and assimilation** | Based on observations, estimates of the source term (location and strength) will be provided. Furthermore, analysis charts are computed by assimilating relevant observations into numerical models. |
| C 5 | **Aviation product development and integration** | Collects the model data from the individual modelling groups, which are produced in C 4, and computes a model ensemble which is passed further to the EUNADICS-AV portal where further output concentration charts and processing for the Air Traffic Management (ATM) is conducted. |
| C 6 | **Data and product delivery** | Distribution of relevant products and visualization of different data sets in a harmonized way. Interface between EUNADICS-AV products and end users. |
| C 7 | **Aviation specific data usage and analysis** | Aviation specific use of EUNADICS-AV products, simulation of air traffic (civil/military perspective), flight management, etc.. |
| C 8 | **User, risk assessment** | Visitor coordination, user survey during the event and collection of feedbacks and additional requirements. |
| C 9 | **Pilots** | Demonstration of a pilot's view. |
| C 10 | **Military/air forces** | Demonstration of military application of EUNADICS-AV data and products. |
| C 11 | **Airports** | Demonstration of an airport's view in case of a disaster event. |
| C 12 | **Airlines** | Demonstration of cost-based decision making in case of a disaster event. |
| C 13 | **Air Navigation Service Providers (ANSPs)** | Role and application of ANSPs in case of a disaster event. |
| C 14 | **EUNADICS-AV Crisis Coordination Cell** | Coordination of the exercise crisis cell and exercise-specific events. |

| C 15 | **National Catastrophic Event Crisis Cell** | Role and applications of national authorities in case of a disaster event. |
|------|------|------|

---

## Author Comment (AC2) · 4 Feb 2020

Dear Andreas! Thanks for your comprehensive revision of the paper and very good suggestions to improve the manuscript. I have uploaded a revised version which contains all your questions and comments (-> yellow, as well as the ones from the first reviewer Tanja Bolic (green)) and our answers and modifications in track change mode. Section 5 has been rewritten according to the reviewers recommendations. Find the file here (somehow I could not upload it for your response, you can also find it when you click on the suplement I have uploaded or the first reviewer): https://edrop.zamg.ac.at/owncloud/index.php/s/AToCeesY2QXFHRo BR Marcus

---

## Referee Report (RR2)

[referee-annotated manuscript omitted]

---

## Author Response (AR2)

[revised manuscript text omitted]

**Kommentiert [m22]:** BOLIC: Isn't this stretching it a bit? It was done for 2 flights out of 98 000...

**Kommentiert [m23R22]:** DONE

**Kommentiert [m24]:** BOLIC: So far so good.

**Kommentiert [m25]:** BOLIC: Add here, after capability - " and all have Safety Risk Assessments that allows flight through/over ask clouds". As that is practically what was done.

**Kommentiert [m26R25]:** DONE

**Kommentiert [m27]:** BOLIC: This is not a valid assumption. With this you are assuming that Flightkeys is the only such product on the market and that all airlines would fly over/across ash. Plenty would fly around in any case, and some would cancel. It would be better to delete this phrase as it opens a can of worms.

**Kommentiert [m28R27]:** DONE

[revised manuscript text omitted]